# Site-specific metal-support interaction to switch the activity of Ir single atoms for oxygen evolution reaction

Jie Wei[1,7], Hua Tang [1,7], Li Sheng[1,7], Ruyang Wang[2], Minghui Fan[1], Jiale Wan[1], Yuheng Wu[1], Zhirong Zhang [1] ✉, Shiming Zhou [1] & Jie Zeng [1,3,4,5,6] ✉

The metal-support interactions (MSI) could greatly determine the electronic properties of single-atom catalysts, thus affecting the catalytic performance. However, the typical approach to regulating MSI usually suffers from interference of the variation of supports or sacrificing the stability of catalysts. Here, we effectively regulate the site-specific MSI of Ir single atoms anchored on Ni layered double hydroxide through an electrochemical deposition strategy. Cathodic deposition drives Ir atoms to locate at three-fold facial center cubic hollow sites with strong MSI, while anodic deposition drives Ir atoms to deposit onto oxygen vacancy sites with weak MSI. The mass activity and intrinsic activity of Ir single-atom catalysts with strong MSI towards oxygen evolution reaction are 19.5 and 5.2 times that with weak MSI, respectively. Mechanism study reveals that the strong MSI between Ir atoms and the support stimulates the activity of Ir sites by inducing the switch of active sites from Ni sites to Ir sites and optimizes the adsorption strength of intermediates, thereby enhancing the activity.

Single-atom catalysts (SACs) have recently drawn extensive attention in the field of energy catalysis due to their excellent catalytic activity originated from the high atomic utilization and unique electronic environment[1–5]. In SACs, the metal atoms individually dispersed on supports directly interact with the supports, maximizing the metal–support interface[6,7]. The metal–support interactions (MSI) could greatly determine the electronic properties via tailoring the charge transfer between metal atoms and supports, thus affecting the catalytic performance of SACs[8,9]. Therefore, tuning MSI is of great importance for developing highly efficient SACs.

The current typical approach to regulating MSI is to modulate the composition and structure of the supports, such as varying the species

of supports and performing the reduction treatment with hydrogen[10,11]. For instance, the strong electron-metal support interactions in $Pt_1/Co_3O_4$ tailored the $5d$ states of Pt atoms, giving rise to a much higher activity than Pt single atoms on $CeO_2$, $ZrO_2$, and graphene in ammonia borane dehydrogenation[11]. In addition, the interactions between Pt atoms and $CoFe_2O_4$ supports could be weakened via a simple reduction treatment, which significantly promoted the activation of the C–H bond towards methane combustion[12]. However, the variation of supports would also affect the catalytic activity, which impeded the comprehensive investigation into the intrinsic reason for the improved catalytic performance. Besides, despite the tailored number of metal−oxygen bonds in the supports after the reduction

[1]Hefei National Research Center for Physical Sciences at the Microscale, University of Science and Technology of China, 230026 Hefei, Anhui, P. R. China. [2]National Synchrotron Radiation Laboratory, University of Science and Technology of China, 230026 Hefei, Anhui, P. R. China. [3]CAS Key Laboratory of Strongly-Coupled Quantum Matter Physics, University of Science and Technology of China, 230026 Hefei, Anhui, P. R. China. [4]Key Laboratory of Surface and Interface Chemistry and Energy Catalysis of Anhui Higher Education Institutes, Department of Chemical Physics, University of Science and Technology of China, 230026 Hefei, Anhui, P. R. China. [5]School of Chemistry & Chemical Engineering, Anhui University of Technology, 243002 Ma'anshan, Anhui, P. R. China. [6]Institute of Advanced Technology, University of Science and Technology of China, 230031 Hefei, Anhui, P. R. China. [7]These authors contributed equally: Jie Wei, Hua Tang, Li Sheng. ✉e-mail: zzhirong@ustc.edu.cn; zengj@ustc.edu.cn

treatment, the stability of catalysts is usually sacrificed, giving rise to the aggregation of single atoms[13,14]. Hence, it is highly desirable to devise a strategy to regulate MSI by simply adjusting the coordination structure of single atoms without altering the supports. Given that the coordination structure of single atoms is strongly related to the specific sites on the surface of supports, the precise regulation of site-specific MSI holds great promise for gaining deep insight into the correlation between MSI and the activity of SACs.

In this work, we effectively regulate the site-specific MSI of Ir SACs by precisely anchoring Ir single atoms on specific sites of Ni layered double hydroxide (Ni LDH) through an electrochemical deposition strategy. Cathodic electrochemical deposition drives isolated Ir atoms to bond with three-fold (3-fold) facial center cubic ($fcc$) hollow sites on Ni LDH (Ir$_1$/Ni LDH-T), while anodic deposition drives Ir atoms connected with the oxygen vacancy site through one apex oxygen atom (Ir$_1$/Ni LDH-V). Compared with Ir$_1$/Ni LDH-V, Ir$_1$/Ni LDH-T possessed more covalent bonds between Ir sites and coordinated oxygen from Ni LDH, thus giving rise to a stronger MSI in Ir$_1$/Ni LDH-T. The impact of different MSI on catalytic performance is explored in oxygen evolution reaction (OER). We find that Ir$_1$/Ni LDH-T exhibits an OER overpotential of $228 \pm 3$ mV at 10 mA cm$^{-2}$, which is 73 mV lower than Ir$_1$/Ni LDH-V. Spectroscopic measurements and density functional theory (DFT) calculations reveal that the strong MSI between Ir atoms and Ni LDH stimulates the activity of Ir sites by inducing the switch of the catalytic active sites from Ni sites to Ir sites and optimizes the adsorption strength of intermediates, thereby enhancing the OER activity.

## Results

### Synthesis and characterizations

The ultrathin Ni LDH was fabricated as the supports using a co-precipitation method[15]. Transmission electron microscopy (TEM) images showed that Ni LDH samples consist of ultrathin nanosheets (Supplementary Fig. 1a). The X-ray diffraction (XRD) pattern of Ni LDH matched well with that of standard β-Ni(OH)$_2$ (JCPDS No. 14-0117) (Supplementary Fig. 1b). The thickness of as-prepared Ni LDH nanosheets was about 3–10 nm measured by atomic force microscopy (AFM; Supplementary Fig. 2). We employed an electrochemical deposition method to selectively anchor Ir single atoms onto specific sites on Ni LDH surface. Ir$_1$/Ni LDH-T and Ir$_1$/Ni LDH-V were prepared through cathodic deposition and anodic deposition with IrCl$_4$ as Ir precursor, respectively (Supplementary Fig. 3). The XRD patterns of both Ir$_1$/Ni LDH-T and Ir$_1$/Ni LDH-V were similar to those of the pristine Ni LDH (Supplementary Figs. 4 and 5). High-resolution transmission electron microscope (HR-TEM) indicated no noticeable metal or metal oxide particles in the two catalysts (Supplementary Fig. 6). The isolated bright spots in the aberration-corrected high-angle annular dark-field scanning TEM (HAADF-STEM) confirmed the atomically dispersed Ir in both Ir$_1$/Ni LDH-T and Ir$_1$/Ni LDH-V (Fig. 1a, b). In addition, energy-dispersive spectroscopic (EDS) elemental mapping results displayed the uniform distribution of Ir atoms across the entire Ni LDH matrix (Fig. 1c, d). The loading mass of Ir could be controlled via changing the concentration of IrCl$_4$ (Supplementary Table 1). To avoid the aggregation of Ir single atoms into clusters at high concentrations of IrCl$_4$ (Supplementary Fig. 7), we set the concentration of IrCl$_4$ as 100 μM. The contents of Ir in Ir$_1$/Ni LDH-T and Ir$_1$/Ni LDH-V were determined to be 2.18 and 2.54 wt% by inductively coupled plasma-atomic emission spectroscopy (ICP-AES), respectively.

The detailed electronic structure and coordination environment of Ir$_1$/Ni LDH-T and Ir$_1$/Ni LDH-V were further revealed by X-ray absorption near-edge spectroscopy (XANES) and extended X-ray absorption fine structure (EXAFS) studies. The normalized Ir $L_3$-edge XANES and EXAFS spectra of Ir$_1$/Ni LDH-T before and after electrooxidation activation confirmed that the Ir−Cl bond was transformed to the Ir−O bond after the OER activation process (Supplementary Fig. 8). The Ir $L_3$-edge XANES spectra of Ir$_1$/Ni LDH-T and Ir$_1$/Ni LDH-V together

with Ir foil and IrO$_2$ as references were shown in Fig. 1e. The intensity of white line for Ir$_1$/Ni LDH-T and Ir$_1$/Ni LDH-V were stronger than that for IrO$_2$, suggesting that the valence states of Ir in both the samples were higher than +4[16]. Besides, the slightly weaker intensity of the white line for Ir$_1$/Ni LDH-T relative to Ir$_1$/Ni LDH-V could be caused by the electrons transfer from Ni LDH to Ir. The Ir $L_3$-edge EXAFS spectra of Ir$_1$/Ni LDH-T and Ir$_1$/Ni LDH-V both exhibited a major peak at ~1.98 Å, which was assigned to the first-shell Ir−O bonding as found in IrO$_2$ (Fig. 1f). The absence of Ir−Ir bonding at around 2.90 Å excluded the formation of Ir-based clusters or nanoparticles, further confirming the single-atom dispersion of Ir in both the samples. By fitting the EXAFS experimental spectra, the first-shell coordination of both Ir$_1$/Ni LDH-T and Ir$_1$/Ni LDH-V was determined to be Ir−O with a coordination number of ~6 (Supplementary Table 2). Moreover, the EXAFS spectra of the two catalysts both discerned a minor peak at around 3.08 Å, which was assigned to the Ir−Ni path from the second-shell coordination. Notably, the coordination number of the Ir−Ni path for Ir$_1$/Ni LDH-T was fitted to be about 2.8, which was larger than that for Ir$_1$/Ni LDH-V (~1.4). This difference was attributed to the different coordination structures of Ir atoms in Ir$_1$/Ni LDH-T and Ir$_1$/Ni LDH-V.

To further elucidate the coordination structure of Ir atoms, we investigated the surface sites of Ni LDH in the process of electrochemical deposition. Under the driving force of electrode potentials, the negative or positive electric field will preferentially induce the deposition of Ir-based cations or Ir-based anions onto the electrode through electrostatic adsorption, respectively. Defective Ni LDH as the support provides different possible sites for anchoring single atoms, including the 3-fold $fcc$ hollow site of oxygen, 3-fold hexagonal close-packed ($hcp$) hollow site of oxygen, and oxygen vacancy site (Supplementary Fig. 9). The surface oxygen atoms of the 3-fold hollow sites provide lone-pair electrons and negative charge to bind Ir-based cations, whereas the deficiency of oxygen induce localized positive charge to combine Ir-based anions. For Ir$_1$/Ni LDH-T, the deposited species in an alkaline environment should be [Ir$^{IV}$Cl$_x$(OH)$_{3-x}$]$^+$ ($1 \le x \le 3$) cations originated from the IrCl$_4$ precursor. The valence state of Ir in the deposited species was determined to be +4 by Ir $L_3$-edge XANES spectra, which showed the similar intensity of white line with IrO$_2$ for Ir$_1$/Ni LDH-T before activation (Supplementary Fig. 8). Moreover, the fitting results for Ir$_1$/Ni LDH-T before activation indicated the first-shell coordination of IrCl$_{1.4}$O$_{4.4}$, since OH$^-$ ions in alkaline environment were prone to attack [IrCl$_3$]$^+$ in the solution or IrCl$_3$O$_3$ on Ni LDH[17]. Besides, the existence of Cl in Ir$_1$/Ni LDH-T was further confirmed by the Cl $2p$ XPS spectra (Supplementary Fig. 10). We further conducted DFT calculations to estimate the formation energy ($\Delta E$) of [IrCl$_3$]$^+$, [IrCl$_2$(OH)]$^+$, and [IrCl(OH)$_2$]$^+$ on the surface sites of Ni LDH (Supplementary Fig. 11). Notably, the $\Delta E$ of [IrCl$_3$]$^+$ on the 3-fold $fcc$ hollow site is −4.97 eV, which is lower than those on the 3-fold $hcp$ hollow site (−3.12 eV) and the oxygen vacancy site (0 eV). Similar results were obtained with [IrCl$_2$(OH)]$^+$ and [IrCl(OH)$_2$]$^+$ as the deposited species, exhibiting the lowest formation energy on the 3-fold $fcc$ hollow site. The higher formation energy of [Ir$^{IV}$Cl$_x$(OH)$_{3-x}$]$^+$ on the $hcp$ site may be caused by electrostatic repulsion between the inner-layer Ni atom and Ir atoms due to the closer distance, leading to lower stability. Therefore, [Ir$^{IV}$Cl$_x$(OH)$_{3-x}$]$^+$ cations were bonded with three oxygen atoms from 3-fold $fcc$ hollow site for Ir$_1$/Ni LDH-T, followed by electrooxidation activation to form IrO$_6$ octahedra linked to Ni LDH. As for Ir$_1$/Ni LDH-V, the negatively charged Ir-based anions were Ir(OH)$_6^{2-}$ with full coordination of IrO$_6$ due to the alkaline environment[18,19]. The $\Delta E$ of Ir(OH)$_6^{2-}$ on the oxygen vacancy site (−3.32 eV) is much lower than those on the 3-fold $fcc$ (0.18 eV) and $hcp$ (0.19 eV) hollow sites, demonstrating the tendency of Ir(OH)$_6^{2-}$ to anchor on the oxygen vacancy site (Supplementary Fig. 12). As such, Ir atoms in Ir$_1$/Ni LDH-V connected with oxygen vacancy sites through one apex oxygen atom. The more covalent bonds between Ir sites and coordinated oxygen in Ir$_1$/Ni LDH-T would give rise to the stronger MSI relative to Ir$_1$/Ni LDH-V.

The XPS measurements for Ir$_1$/Ni LDH-T and Ir$_1$/Ni LDH-V were performed to further clarify the MSI between Ir and Ni LDH. As displayed in Fig. 1h, compared with Ni LDH, the Ni 2$p$ peaks of Ir$_1$/Ni LDH-T shifted to higher binding energy, indicating an increased valence state of Ni. With regard to Ir$_1$/Ni LDH-V, the Ni 2$p$ peaks showed no obvious change after anchoring Ir single atoms. Besides, the Ir 4$d$ XPS spectra of Ir$_1$/Ni LDH-T exhibited a slight negative shift compared with Ir$_1$/Ni LDH-V (Supplementary Fig. 13), suggesting a slightly lower valence state of Ir single atoms in Ir$_1$/Ni LDH-T, which was consistent with the results of XANES. Furthermore, cyclic voltammetry (CV) curves showed that the oxidation peak of Ni from +2 to +3 in Ir$_1$/Ni LDH-T shifted lower by 20 mV compared with Ni LDH whereas the oxidation peak in Ir$_1$/Ni LDH-V maintained the same potential with Ni LDH (Fig. 1i). As such, the stronger MSI between Ir atoms and Ni LDH in Ir$_1$/Ni LDH-T enhanced the electron transfer from Ni to Ir, resulting in a higher oxidation state of Ni in Ir$_1$/Ni LDH-T.

## Electrocatalytic performance towards oxygen evolution reaction

To explore the influence of MSI on the catalytic performance of Ir-based SACs, the as-prepared Ni LDH, Ir$_1$/Ni LDH-V, and Ir$_1$/Ni LDH-T were applied as electrocatalysts towards OER, one of the most essential reactions for clean energy conversion and storage application[20-28]. The linear sweep voltammetry (LSV) curves (Supplementary Figs. 14 and 15) measured in 1.0 M KOH solution show that Ir$_1$/Ni LDH-T exhibited a remarkable activity with an overpotential ($\eta$) of 228 ± 3 mV at 10 mA cm$^{-2}$, which significantly outperformed Ir$_1$/Ni LDH-V (301 ± 8 mV) and Ni LDH (321 ± 3 mV). The overpotentials of Ir$_1$/Ni LDH-T at 50 and 100 mA cm$^{-2}$ were 276 and 289 mV, respectively, which were also superior to those of Ir$_1$/Ni LDH-V and Ni LDH. Moreover, Ir$_1$/Ni LDH-T achieved a mass activity of 0.33 A mg$^{-1}$ at the overpotential of 300 mV, which was 19.5 times higher than Ir$_1$/Ni LDH-V (Supplementary Fig. 16a, b). In addition, the normalized mass activity of Ir$_1$/Ni LDH-T based on the content of Ir loading at the overpotential of 300 mV reached up to 15.2 A mg$_{Ir}^{-1}$, which was 22.7-fold higher than that of Ir$_1$/Ni LDH-V (Supplementary Fig. 16c, d). Based on the number of overall metals, the calculated turnover frequency (TOF) (Supplementary Fig. 17a) of Ir$_1$/Ni LDH-T was about 0.12 s$^{-1}$ at the overpotential of 300 mV, which was much higher than those of Ir$_1$/Ni LDH-V (0.006 s$^{-1}$) and Ni LDH (0.003 s$^{-1}$). Besides, the TOF based on the number of Ir sites of Ir$_1$/Ni LDH-T was estimated to be 7.6 s$^{-1}$, largely surpassing Ir$_1$/Ni LDH-V by 19.0 times (Supplementary Fig. 17b). In

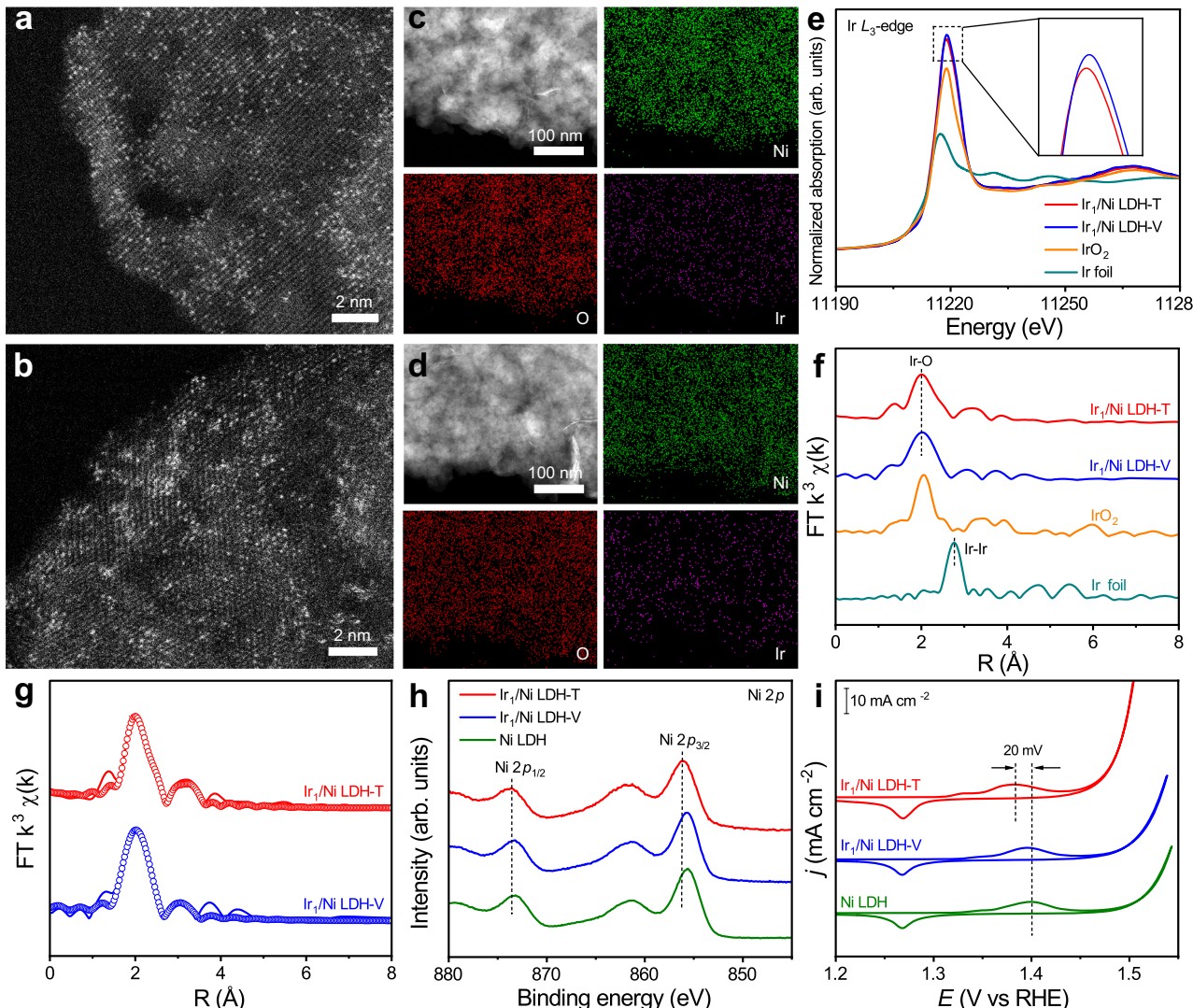

**Fig. 1 | Structural characterizations of Ir$_1$/Ni LDH-T and Ir$_1$/Ni LDH-V.** **a**, **b** HAADF-STEM images of Ir$_1$/Ni LDH-T (**a**) and Ir$_1$/Ni LDH-V (**b**). **c**, **d** EDS elemental mapping of Ir$_1$/Ni LDH-T (**c**) and Ir$_1$/Ni LDH-V (**d**). **e**, **f** Normalized XANES (**e**) and EXAFS (**f**) spectra at the Ir $L_3$-edge for Ir$_1$/Ni LDH-T and Ir$_1$/Ni LDH-V. Ir foil and IrO$_2$ were used as references. **g** Experimental and fitting EXAFS results of Ir$_1$/Ni LDH-T and Ir$_1$/Ni LDH-V. The experimental and fitting results were shown with solid lines and circles, respectively. **h** Ni 2$p$ XPS spectra of Ir$_1$/Ni LDH-T, Ir$_1$/Ni LDH-V, and Ni LDH. **i** CV curves of Ir$_1$/Ni LDH-T, Ir$_1$/Ni LDH-V, and Ni LDH.

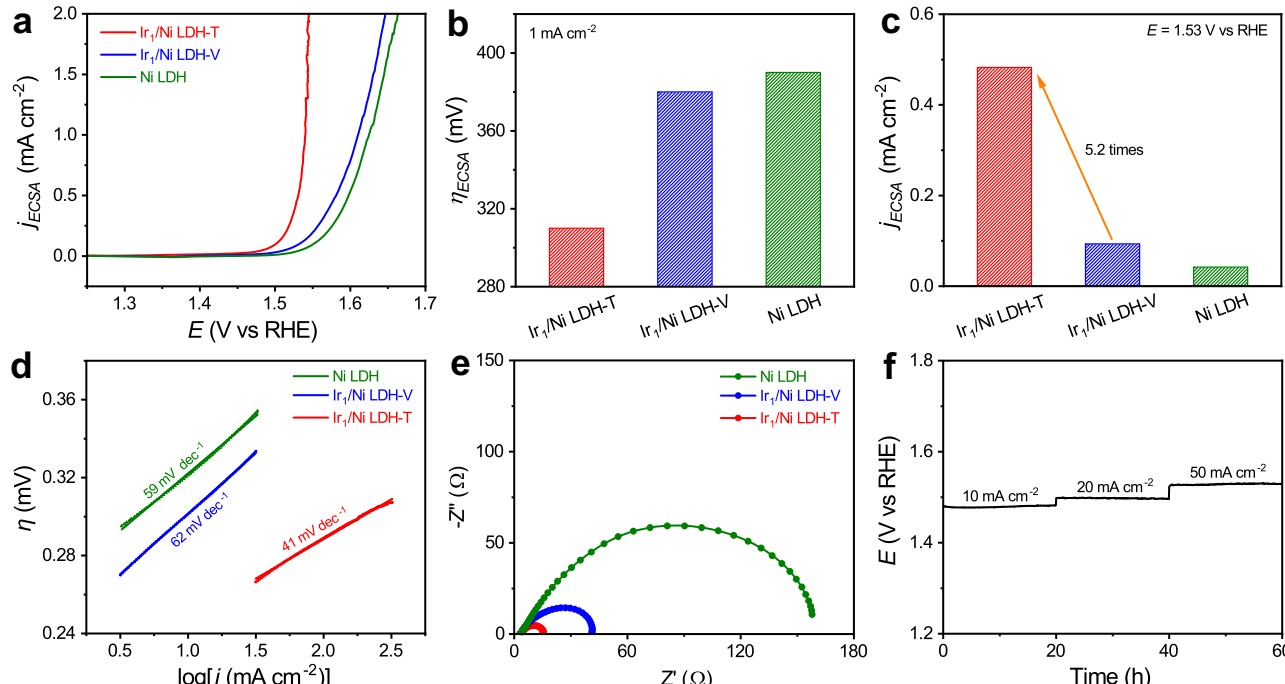

**Fig. 2 | Electrocatalytic performance towards OER. a** ECSA-normalized polarization curves of Ni LDH, Ir$_1$/Ni LDH-V, and Ir$_1$/Ni LDH-T in 1.0 M KOH. **b** Overpotentials of different catalysts at the current density of 1 mA cm$^{-2}$, respectively. **c** The ECSA-normalized current density at an overpotential of 300 mV for Ir$_1$/Ni LDH-T, Ir$_1$/Ni LDH-V, and Ni LDH, respectively. **d** Tafel slopes of Ni LDH, Ir$_1$/Ni LDH-V, and Ir$_1$/Ni LDH-T. **e** Electrochemical impedance spectra of Ni LDH, Ir$_1$/Ni LDH-V, and Ir$_1$/Ni LDH-T at 1.57 V vs. RHE. The $R_{ct}$ values for Ni LDH, Ir$_1$/Ni LDH-V, and Ir$_1$/Ni LDH-T were 90.7 ± 1.5, 26.6 ± 0.1, and 8.2 ± 0.2 Ω, respectively. **f** Chronopotentiometric curves of Ir$_1$/Ni LDH-T towards OER at 10, 20, and 50 mA cm$^{-2}$ for 20 h, respectively.

addition, the activity of Ni LDH after the cathodic and anodic treatment without Ir precursor remained almost unchanged, suggesting that the enhanced performance of Ir$_1$/Ni LDH-T was attributed to the Ir center with site-specific MSI rather than Ni LDH (Supplementary Fig. 18).

To clarify the intrinsic activity of the samples, we tested electrochemical double-layer capacitance ($C_{dl}$) to assess electrochemically active surface area (ECSA) (Supplementary Fig. 19). The $C_{dl}$ value of Ir$_1$/Ni LDH-T was determined to be 23.4 mF cm$^{-2}$, which was higher than those of Ir$_1$/Ni LDH-V (6.2 mF cm$^{-2}$) and Ni LDH (6.0 mF cm$^{-2}$), indicating an increment of ECSA. As shown in Fig. 2a, the ECSA-normalized OER activity still followed the order of Ir$_1$/Ni LDH-T > Ir$_1$/Ni LDH-V > Ni LDH, suggesting that the OER activity of Ir$_1$/Ni LDH-T was also intrinsically improved. Specially, the overpotential of Ir$_1$/Ni LDH-T at 1 mA cm$^{-2}$ was only 310 mV, which was lower than that of Ir$_1$/Ni LDH-V (380 mV) and Ni LDH (390 mV) (Fig. 2b). Besides, the ECSA-normalized current density for Ir$_1$/Ni LDH-T was estimated to be 0.48 mA cm$^{-2}$ at the overpotential of 300 mV, which was about 5.2-fold and 11.4-fold higher than those for Ir$_1$/Ni LDH-V (0.093 mA cm$^{-2}$) and Ni LDH (0.042 mA cm$^{-2}$) (Fig. 2c). Figure 2d displays the Tafel plots for Ir$_1$/Ni LDH-T, Ir$_1$/Ni LDH-V, and Ni LDH, where their Tafel slopes were determined to be 41, 62, and 59 mV dec$^{-1}$, respectively. The lowest Tafel slope indicated the fastest reaction kinetics for Ir$_1$/Ni LDH-T. Electrochemical impedance spectroscopy (EIS) measurements were further conducted to probe the interfacial charge-transfer resistances of the catalysts. The smallest semicircle diameter of Ir$_1$/Ni LDH-T among the three catalysts suggested its fastest charge transfer at the interface (Fig. 2e). Durability tests of Ir$_1$/Ni LDH-T with the superior activity were carried out via chronopotentiometry test at the current density of 10, 20, and 50 mA cm$^{-2}$, respectively. As plotted in Fig. 2f, Ir$_1$/Ni LDH-T kept stable for 20-h operation. The percentages of leached Ir and Ni after the durability test at 10, 20, and 50 mA cm$^{-2}$ for 20 h were all <5% detected by inductively coupled plasma-mass spectrometry

(ICP-MS) (Supplementary Fig. 20). After the durability test, the atomic dispersion of Ir atoms was maintained without any noticeable aggregation as shown in the HAADF-STEM image and EDS elemental mapping result (Supplementary Fig. 21a, b). Moreover, the XRD pattern and XPS spectra of Ir$_1$/Ni LDH-T after the durability test also exhibited no observable change, further implying the structural robustness of Ir$_1$/Ni LDH-T (Supplementary Fig. 21c, d).

**In situ spectroscopic characterizations**

To gain an in-depth understanding of the reaction mechanism, we conducted a series of in situ Raman experiments using the homemade in situ Raman cell (Supplementary Fig. 22). The in situ Raman spectra of Ir$_1$/Ni LDH-T, Ir$_1$/Ni LDH-V, and Ni LDH were collected over the potentials ranging from 1.2 to 1.6 V vs. reversible hydrogen electrode (RHE), respectively (Supplementary Fig. 23). During the positive potential excursion, two bands for the three catalysts gradually emerged at around 481 and 562 cm$^{-1}$, which were assigned to the E$_g$ bending vibration ($\delta$(Ni$^{3+}$–O)) and A$_{1g}$ stretching vibration ($\nu$(Ni$^{3+}$–O)) mode of NiOOH, respectively[29,30]. However, compared with Ir$_1$/Ni LDH-V and Ni LDH, the peak of $\delta$(Ni$^{3+}$-O) at 1.6 V vs. RHE for Ir$_1$/Ni LDH-T shifted to higher wavenumbers from 475 to 481 cm$^{-1}$, implying that the MSI between single-atom Ir and Ni LDH was stronger in Ir$_1$/Ni LDH-T (Fig. 3a). We further carried out in situ Raman measurements with a potential interval of 0.01 V to confirm such a change of Ni$^{2+}$/Ni$^{3+}$ electrochemical redox during OER process. The emergence of the Raman peaks for Ni$^{3+}$−O occurred at 1.35, 1.37, and 1.37 V vs. RHE for Ir$_1$/Ni LDH-T, Ir$_1$/Ni LDH-V, and Ni LDH, respectively (Fig. 3b, c and Supplementary Fig. 24). The redox process of Ni$^{2+}$/Ni$^{3+}$ for Ir$_1$/Ni LDH-T started at a more negative potential, which was attributed to the stronger MSI in Ir$_1$/Ni LDH-T.

To confirm the real active centers during the OER process, we conducted oxygen-isotope-labeling in situ Raman spectra for Ir$_1$/Ni LDH-T, Ir$_1$/Ni LDH-V, and Ni LDH. The oxygen atoms of catalysts

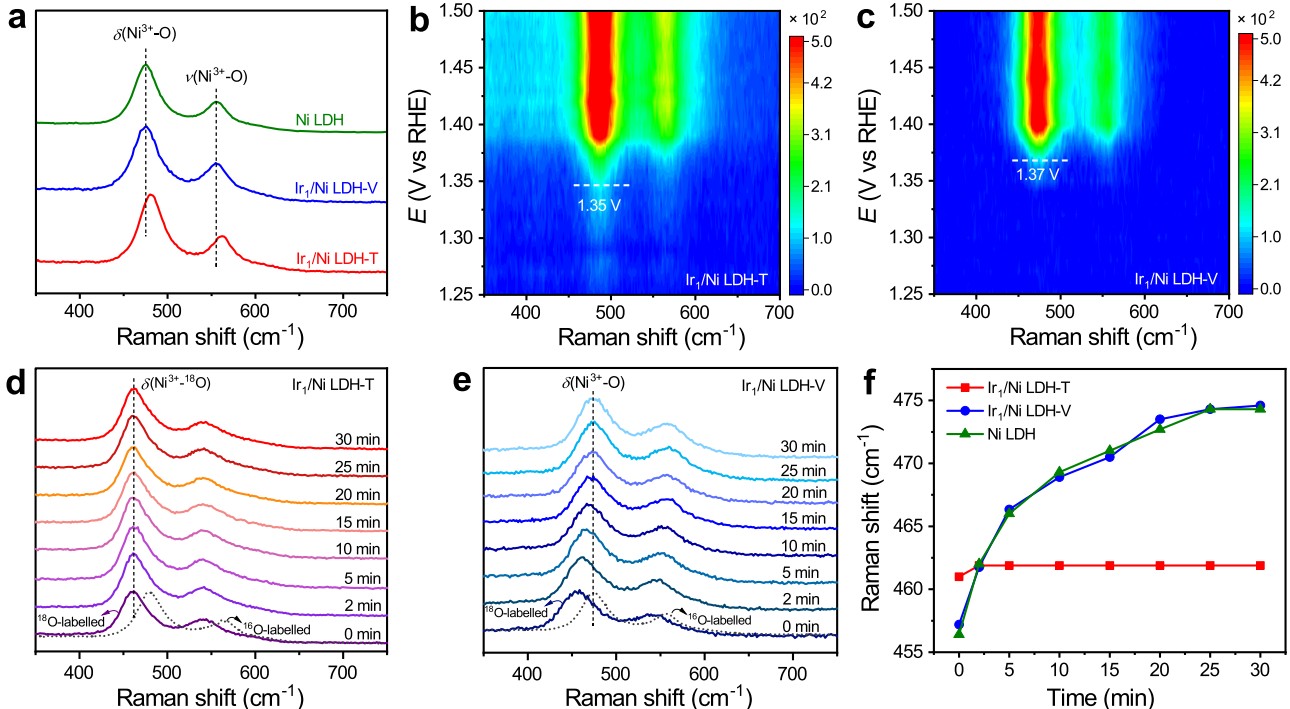

**Fig. 3 | In situ electrochemical Raman spectra. a** In situ Raman spectra of Ni LDH, $Ir_1$/Ni LDH-V, and $Ir_1$/Ni LDH-T at 1.6 V vs. RHE, respectively. **b, c** In situ Raman spectra of $Ir_1$/Ni LDH-T (**b**) and $Ir_1$/Ni LDH-V (**c**) during the linear sweep voltammetry measurement, in which the dash lines mark the conversion potential from $Ni^{2+}$ to $Ni^{3+}$. **d, e** In situ Raman spectra of $^{18}O$-labeled $Ir_1$/Ni LDH-T (**d**) and $Ir_1$/Ni LDH-V (**e**) acquired at 50 mA cm$^{-2}$ in 1.0 M KOH–$H_2^{16}O$ solution, respectively. **f** Shifts of Raman bands of $Ni^{3+}$–O with the increase of time during the oxygen isotope exchange experiments.

were initially labeled with $^{18}O$ by applying LSV measurement from 0.50 to 1.65 V vs. RHE for five cycles in 1.0 M KOH–$H_2^{18}O$ solution with the addition of 50 mM $KClO_4$ as an internal standard. As shown in Supplementary Fig. 25, compared with $^{16}O$-labeled NiOOH, the peak of $ClO_4^-$ remained unchanged whereas the peak of $\delta(Ni^{3+}$–O) for $^{18}O$-labeled NiOOH shifted to lower wavenumbers by about 20 cm$^{-1}$, manifesting the occurrence of isotope exchange. After the quick replacement of the electrolyte with 1.0 M KOH–$H_2^{16}O$ solution using a creep pump, the catalysts labeled with $^{18}O$ were subjected to OER at 50 mA cm$^{-2}$. As the reaction time increased, the peaks of $\delta(Ni^{3+}$–O) and $\nu(Ni^{3+}$–O) for $Ir_1$/Ni LDH-V and Ni LDH gradually shifted back to the positions observed for the $^{16}O$-labeled samples (Fig. 3e and Supplementary Fig. 26). The results suggest that the $^{18}O$-labeled oxygen in $Ir_1$/Ni LDH-V and Ni LDH was easily exchanged with the $^{16}O$ atom in $^{16}OH^-$ during the OER process, indicating that Ni could be the active sites for $Ir_1$/Ni LDH-V and Ni LDH[15]. Nevertheless, in the case of $Ir_1$/Ni LDH-T, the peaks of $\delta(Ni^{3+}$–O) and $\nu(Ni^{3+}$–O) exhibited no discernible shift, suggesting that the dominant active sites for $Ir_1$/Ni LDH-T could be Ir single atoms rather than Ni species (Fig. 3f). Therefore, the integration of Ir on Ni LDH with strong MSI could induce the switch of the active sites from Ni sites to Ir sites.

## DFT calculations

To get a theoretical insight into the reaction mechanism, we employed the DFT calculation to investigate the reaction process[31,32]. The structure models of $Ir_1$/Ni LDH-T and $Ir_1$/Ni LDH-V were built on two-layer 3 × 3 × 3 supercells based on the structure details discussed above (Fig. 4a, b). The results of charge density differences indicated that the electronic density of Ir atoms in $Ir_1$/Ni LDH-T was higher than that in $Ir_1$/Ni LDH-V (Fig. 4c, d). The Bader charge of Ir in $Ir_1$/Ni LDH-T is 1.75, lower than that in $Ir_1$/Ni LDH-V (2.02), indicating a lower oxidation state of Ir in $Ir_1$/Ni LDH-T. These results verified that the MSI induced by $Ir_1$/Ni LDH-T was stronger than that induced by $Ir_1$/Ni LDH-V. Then we

calculated the projected density of states (PDOS) to elucidate the differences in electronic interactions between $Ir_1$/Ni LDH-T and $Ir_1$/Ni LDH-V (Fig. 4e, f and Supplementary Fig. 27). The positions of the $d$-band center for $Ir_1$/Ni LDH-T, $Ir_1$/Ni LDH-V, and Ni LDH were −1.59, −1.64, and −1.52 eV, respectively. The moderate $d$-band center of $Ir_1$/Ni LDH-T is beneficial for balancing the adsorption and desorption abilities of intermediates (*OH, *O, and *OOH), thereby boosting the OER performance[32–35].

Furthermore, we calculated the free energy diagram for OER based on the conventional adsorbate evolution mechanism (AEM), which consists of four concurrent proton-electron transfer steps[31]. As shown in Supplementary Figs. 28 and 29, the step from *OH to *O over NiOOH exhibited the highest free energy change ($\Delta G$) of 2.20 eV among all the steps, which served as the rate-determining step (RDS). For $Ir_1$/Ni LDH-V, the RDS on Ir sites was determined to be the formation of *OH with an $\Delta G$ of 2.34 eV, which was higher than that of RDS for NiOOH (Fig. 4g–i). In this case, the OER process was more easily proceeded on Ni sites rather than Ir sites for $Ir_1$/Ni LDH-V. In contrast, the energy barrier of RDS (the step from *O to *OOH) on Ir sites for $Ir_1$/Ni LDH-T was 1.77 eV, which was significantly lower than that for NiOOH, indicating that Ir atoms preferentially served as the true active sites for $Ir_1$/Ni LDH-T (Fig. 4g–i). Accordingly, the strong MSI between Ir and Ni LDH optimized the adsorption energy for the oxygen-containing intermediates, thereby facilitating the switch of the active sites from Ni sites to Ir sites.

## Discussion

In summary, we reported Ir SACs with site-specific MSI by a facile electrochemical deposition technique. Compared with $Ir_1$/Ni LDH-V, the stronger MSI in $Ir_1$/Ni LDH-T was reflected by more covalent bonds between Ir sites and coordinated oxygen from Ni LDH. The $Ir_1$/Ni LDH-T exhibited superior OER activity, while $Ir_1$/Ni LDH-V showed little improvement in activity compared with pristine Ni LDH. The strong MSI between Ir single atoms and Ni LDH in $Ir_1$/Ni LDH-T not only

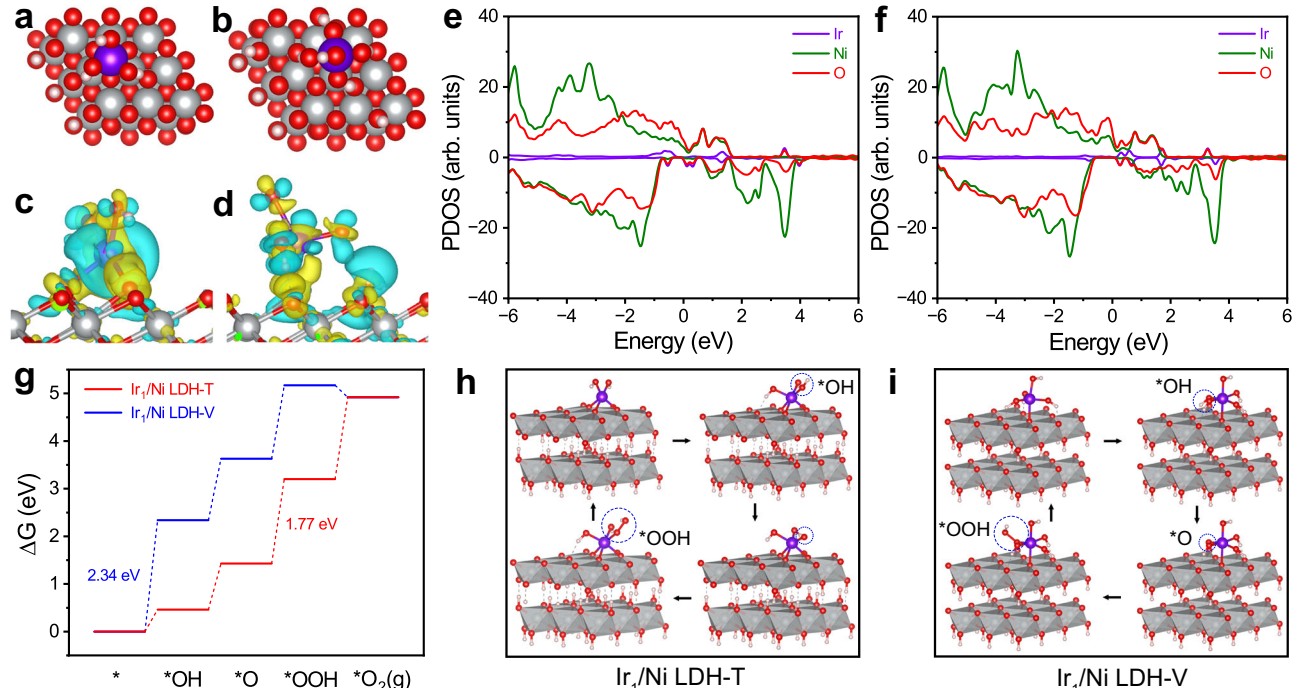

**Fig. 4 | OER mechanism studies for Ir₁/Ni LDH-T and Ir₁/Ni LDH-V. a, b** Schematic structure models of Ir₁/Ni LDH-T (**a**) and Ir₁/Ni LDH-V (**b**) from top views. **c, d** Charge density differences of Ir atoms on Ir₁/Ni LDH-T (**c**) and Ir₁/Ni LDH-V (**d**). The yellow and cyan areas indicate electron accumulation and depletion, respectively. **e, f** PDOS in Ir₁/Ni LDH-T (**e**) and Ir₁/Ni LDH-V (**f**). **g** Free energy diagrams of Ir₁/Ni LDH-T and Ir₁/Ni LDH-V with Ir as the active sites. **h, i** The schematic OER pathway for Ir₁/Ni LDH-T (**h**) and Ir₁/Ni LDH-V (**i**). The pink, red, gray, and purple spheres represent H, O, Ni, and Ir atoms, respectively. The reaction intermediates are indicated by blue circles.

optimized the adsorption strength of oxygen-containing intermediates but also induced the switch of the catalytic active sites from Ni sites to Ir sites, thereby boosting the catalytic performance. This work provides a novel strategy to design highly active catalysts with site-specific MSI by electrochemically selective anchoring of single atoms and offers an in-depth understanding of the correlation between MSI and OER performance.

## Methods

### Chemicals
Nickel nitrate (Ni(NO₃)₂·6H₂O), sodium nitrate (NaNO₃), sodium hydroxide (NaOH), and formamide were purchased from Shanghai Chemical Reagent Company. Iridium (IV) chloride hydrate (IrCl₄·xH₂O, Ir ≥ 56.0%) was purchased from Aladdin. Potassium hydroxide (99.99%) was purchased from Macklin. Nafion solution (5 wt%) was purchased from Sigma-Aldrich. Water–¹⁸O (97 atom% ¹⁸O) was purchased from Beijing InnoChem Science & Technology Co., Ltd. All the other chemicals were of analytical grade and used as received without further purification. All aqueous solutions were prepared using deionized water with a resistivity of 18.2 MΩ cm⁻¹.

### Synthesis of Ni LDH
Ni LDH nanosheets were fabricated according to a previous report[15]. A solution of 10 mM NaNO₃ (20 mL) containing 23 vol% formamide was heated to 80 °C under stirring. Then, 10 mL solution of 37.5 mM Ni(NO₃)₂·6H₂O and 0.25 M NaOH were simultaneously added dropwise to maintain the solution pH at -10. The resulting solution was cooled to room temperature under nitrogen gas bubbling. Then the precipitate was washed with a mixture of ethanol and deionized water (1:1 vol%) at least three times. Finally, the precipitate was dispersed in a small amount of water and lyophilized.

### Synthesis of Ir₁/Ni LDH-T and Ir₁/Ni LDH-V
Ir₁/Ni LDH-T and Ir₁/Ni LDH-V were prepared using our previously reported electrochemical deposition method[19]. The deposition was performed in a three-electrode system (CHI 660E, Shanghai CH Instruments), with a Ni foam (3 × 3 mm), a carbon rod, and an Ag/AgCl electrode as the working, counter, and reference electrodes, respectively. 2.5 mg of Ni LDH were dispersed in 0.6 mL of ethanol, 0.6 mL of water, and 50 μL of Nafion by sonication for 30 min to form a homogeneous mixture. Then 25 μL of the mixture was loaded onto the working electrode. To synthesize Ir₁/Ni LDH-T, 10 cycles of linear sweep voltammetry at the applied potentials ranging from 0.1 to −0.4 V vs. RHE were employed for the pretreatment of the working electrode. Afterward, 100 μM IrCl₄ as the Ir precursor was added into the electrolyte, followed by stirring for 10 min. Then cathodic deposition was performed by scanning the linear sweep voltammetry at the potentials from 0.1 to −0.4 V vs. RHE for 10 cycles, followed by a CV method under potentials ranging from 1.1 to 1.8 V vs. RHE. The scan rate was set at 5 mV s⁻¹. In the case of Ir₁/Ni LDH-V, 3 cycles of the CV method at the potentials ranging from 1.1 to 1.8 V vs. RHE were applied for the pretreatment of the working electrode. Afterward, 100 μM IrCl₄ as the Ir precursor was added to the electrolyte, followed by stirring for 10 min. The anodic deposition was then conducted by scanning the linear sweep voltammetry at the potentials from 1.1 to 1.8 V vs. RHE for 5 cycles. The scan rate was set at 5 mV s⁻¹. The obtained samples were finally washed with water, followed by being dried for the electrochemical measurements. All potentials were measured against the Ag/AgCl electrode and converted to the RHE reference scale by $E$ (V vs. RHE) = $E$ (V vs. Ag/AgCl) + 0.194 V + 0.0591 pHV.

### Catalyst characterization
TEM images were obtained on a transmission electron microscope (Hitachi H-7650) operating at an acceleration voltage of 100 kV. AFM

was measured on the Veeco DI Nanoscope MultiMode V system. XRD patterns were collected using an X-ray diffractometer (Philips X'Pert Pro Super) with Cu-Kα radiation ($\lambda = 1.54178\,Å$). HAADF-STEM images and EDS elemental mapping were obtained on a field-emission transmission electron microscope (JEOL ARM-200F) operating at an accelerating voltage of 200 kV using Cu-based TEM grids. XPS measurements were conducted on an X-ray photoelectron spectrometer (VG ESCALAB MK II) with MgKα = 1253.6 eV as the exciting source. ICP-AES (Atomscan Advantage, Thermo Jarrell Ash) were applied for the measurement of concentrations for metal species. The prepared electrode was rinsed several times with ultrapure water and then subjected to ultrasonication in ethanol. The catalyst collected in ethanol was washed three times with ethanol and water by centrifugation. The resulting solid was freeze-dried in a vacuum freeze-dryer overnight. Before analysis using ICP-AES, a proper amount of powder sample was treated by hot aqua regia to get a clear solution. Line 221.6 and 224.2 were selected for Ni and Ir detection, respectively. ICP-MS (Thermo Fisher iCAP RQ) analysis was performed using $^{115}In$ and $^{73}Ge$ as internal standards with a concentration of 40 ppb.

XANES and EXAFS spectra at the Ir $L_3$-edge were recorded at the BL14W1, Shanghai Synchrotron Radiation Facility, China. The energy was calibrated by Ir foil. Ir foil and $IrO_2$ were used as reference samples and measured in the transmission mode. The samples were measured in fluorescence mode. The Athena software package was used to analyze the data.

### Electrochemical measurements

To assess the electrocatalytic activity of the samples, an electrochemical workstation (CHI 660E, Shanghai CH Instruments) was employed. All electrochemical tests were carried out in a standard three-electrode system under ambient conditions in 1.0 M KOH electrolyte. The content of Fe in 1.0 M KOH was quantified to be ~88 ppb using ICP-MS. The working electrode was a $0.3 \times 0.3\,cm^2$ Ni foam electrode and the loading of catalysts was $0.55\,mg\,cm^{-2}$. A carbon rod and an Ag/AgCl electrode served as the counter and reference electrodes, respectively. The polarization curves of catalysts were measured by scanning linear sweep voltammetry under the potentials ranging from 1.1 to 1.8 V vs. RHE with a scan rate of $5\,mV\,s^{-1}$. To correct for solution resistance, the potentials were adjusted using the equation $E_{iR\text{-corrected}} = E$ (V vs. RHE) $- iR_u$, where $i$ and $R_u$ represented the current and the uncompensated ohmic electrolyte resistance, respectively. $R_u$ was extracted from the Nyquist plot through alternating current impedance in the region of high frequency[36]. The $R_u$ value was measured to be ~3.0 Ω (1.0 M KOH). The ECSA of the catalyst was obtained by the following equation: ECSA = $R_f*S$, where $R_f$ and $S$ represented the roughness factor and the surface area of the glassy carbon electrode ($0.09\,cm^2$ in this case), respectively. $R_f$ could be calculated according to $R_f = C_{dl}/60$ based on the $C_{dl}$ of an ideal flat oxide surface ($60\,\mu F\,cm^{-2}$). To obtain the $C_{dl}$ of the samples, we conducted the CV measurements from 1.06 to 1.16 V vs. RHE at different scan rates (20, 40, 60, 80, and $100\,mV\,s^{-1}$). $C_{dl}$ was estimated by plotting the $\Delta j$ ($\Delta j = j_a - j_c$) at 1.11 V vs. RHE against the scan rates. The $j_a$ and $j_c$ represented the anodic and cathodic current density, respectively. The linear slope was equivalent to twice of $C_{dl}$[16]. To evaluate the specific activities and mass activities of the samples, the current densities were normalized based on ECSA and mass loading of catalysts, respectively. The TOFs were estimated using the following equation: TOF = $(j \times A)/(4 \times F \times m)$, where $j$ represented the current density, $A$ was the surface area of the electrode, $F$ represented the Faraday constant, and $m$ represented the mole number of metals on the electrode. EIS measurements were carried out in a standard three-electrode system at room temperature in 1.0 M KOH electrolyte at a potential of 1.57 V vs. RHE with a sinusoidal wave amplitude of 5 mV and a frequency scan range of 100 kHz–0.05 Hz. The working electrode was a $0.3 \times 0.3\,cm^2$ Ni foam electrode and the loading of catalysts was $0.55\,mg\,cm^{-2}$. A carbon rod and an Ag/AgCl electrode served as the counter and reference electrodes, respectively. To perform the durability test, the galvanostatic mode was used in 1.0 M KOH electrolyte at room temperature with current densities of 10, 20, and $50\,mA\,cm^{-2}$, respectively.

### In situ Raman spectroscopy

The Raman spectra were carried out on a confocal microscope Raman system (Horiba LabRAM HR Evolution). The excitation wavelength was 532 nm, and a ×50 microscope objective with a numerical aperture of 0.55 was used in all Raman measurements. Before measurements, the Raman shift range was calibrated using the $520.6 \pm 0.5\,cm^{-1}$ peak of silicon. A homemade Teflon Raman cell was used for the in situ electrochemical Raman measurements with a 3 mm Au electrode, a Pt wire, and an Ag/AgCl electrode as the working, counter, and reference electrodes, respectively. The working electrode was prepared by casting 5 μL ink on a 3 mm Au electrode. $Ir_1$/Ni LDH-T and $Ir_1$/Ni LDH-V were synthesized by the same electrochemical deposition method. The potential was controlled by an electrochemical workstation (CHI 660E, Shanghai CH Instruments). To label the catalyst with oxygen-18, we conducted LSV measurements in a 1.0 M KOH–$H_2^{18}O$ solution within the potential range of 0.50–1.65 V vs. RHE for five cycles.

### DFT calculation method

All the spin-unrestricted DFT calculations with Hubbard U (DFT + U) corrections were performed using the Vienna ab initio simulation package (VASP). For the description of exchange-correlation effects, the Perdew–Burke–Ernzerhof function within the generalized gradient approximation was employed[37]. A plane wave basis set was generated with a kinetic energy cutoff of 450 eV, and $k$-points were sampled using a $3 \times 3 \times 1$ and $5 \times 5 \times 1$ Gamma mesh for geometry optimization and electronic structure calculations of all intermediates on NiOOH (001), $Ir_1$/Ni LDH-T, and $Ir_1$/Ni LDH-V, respectively. For a better description of the 3$d$ electrons, the Hubbard effective term $U_{eff} = 4.4$ eV for Ni was added to the PBE functional through the rotationally invariant approach proposed by Dudarev and co-workers[38]. A $3 \times 3$ single-cell NiOOH (001) surface with double atomic layers consisted of 36 oxygen atoms, 18 Ni atoms, and 18 H atoms. A vacuum layer of 15 Å in the z-direction was set to avoid unwanted interactions between periodic images. The convergence of energy and force were set to be $10^{-5}$ eV and 0.03 eV/Å, respectively. The DFT-D3 method was adopted to take van der Waals correction into consideration[39].

### Theoretical evaluation of OER activity

The theoretical overpotentials for NiOOH, $Ir_1$/Ni LDH-T, and $Ir_1$/Ni LDH-V interfaces were determined by following the conventional four-step electron–proton-coupled OER mechanism[40,41]. The OER process generally involves the following steps:

$$H_2O(l) + * \rightarrow {}^*OH + H^+ + e^- \tag{1}$$

$$^*OH \rightarrow {}^*O + H^+ + e^- \tag{2}$$

$$^*O + H_2O(l) \rightarrow {}^*OOH + H^+ + e^- \tag{3}$$

$$^*OOH \rightarrow O_2(g) + H^+ + e^- \tag{4}$$

where * represented the active site on the model of a specific surface, (l) and (g) represented the liquid and gas phases, respectively. Considering that the standard thermodynamic potential was 1.23 V for water oxidation, the Gibbs free energy for forming one water molecule was fixed to the experimental value $\Delta G_{H_2O} = -2 \times 1.23 = -2.46$ eV. In this case, the calculation for bond energy of $O_2$ could be avoided, which was inaccurately interpreted by DFT[42]. The free energy of the $O_2$

molecule was calculated as

$$G_{O_2} = 2\,G_{H_2O} - 2\,G_{H_2} - 2\,\Delta G_{H_2O} \tag{5}$$

where $G_{H_2O}$ and $G_{H_2}$ represented the calculated free energies of $H_2O$ and $H_2$, respectively. To calculate the Gibbs free energy for the reaction intermediates, we applied the computational hydrogen electrode (CHE) model[43]. The difference in Gibbs free energies of these intermediates was calculated as $\Delta G = \Delta E_{DFT} + \Delta(ZPE - TS)$, where $\Delta E_{DFT}$, ZPE, $T$, and $S$ represented adsorption enthalpy, zero-point energy, temperature, and entropy, respectively. The change of Gibbs energy for intermediates (*O, *OH, and *OOH) and every OER step can be calculated as follows[44]:

$$\Delta G_* = 0 \tag{6}$$

$$\Delta G_{*OH} = G_{*OH} - G_* + 0.5 \times G_{H_2} - G_{H_2O} - eU \tag{7}$$

$$\Delta G_{*O} = G_{*O} - G_* + G_{H_2} - G_{H_2O} - 2eU \tag{8}$$

$$\Delta G_{*OOH} = G_{*OOH} - G_* + 1.5 \times G_{H_2} - 2 \times G_{H_2O} - 3eU \tag{9}$$

$$\Delta G_{*+O_2} = 4 \times 1.23\,\text{eV} - 4eU \tag{10}$$

$$\Delta G_1 = \Delta G_{*OH} \tag{11}$$

$$\Delta G_2 = \Delta G_{*O} - \Delta G_{*OH} \tag{12}$$

$$\Delta G_3 = \Delta G_{*OOH} - \Delta G_{*O} \tag{13}$$

$$\Delta G_4 = 4.92 - \Delta G_{*OOH} \tag{14}$$

where $U$ was the potential against normal hydrogen electrode (NHE) at standard conditions ($T = 298.15\,\text{K}$, $P = 1\,\text{bar}$, pH = 0).

## Data availability

The experiment data that support the findings of this study are available from the corresponding author upon reasonable request. The source data underlying Figs. 1–4 and Supplementary Figs. 1–29 are provided as a Source Data file. Source data are provided with this paper.

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

## Acknowledgements

This work was supported by the National Key Research and Development Program of China (2021YFA1500500 and 2019YFA0405600), CAS Project for Young Scientists in Basic Research (YSBR-051), National Science Fund for Distinguished Young Scholars (21925204), NSFC (U19A2015, 22221003, 22250007, and 22302184), Fundamental Research Funds for the Central Universities, K.C. Wong Education (GJTD-2020-15), Collaborative Innovation Program of Hefei Science Center, CAS (2022HSC-CIP004), the Joint Fund of the Yulin University and the Dalian National Laboratory for Clean Energy (YLU-DNL Fund 2022012), the DNL Cooperation Fund, CAS (DNL202003), International Partnership Program of Chinese Academy of Sciences (123GJHZ2022101GC), the Anhui Natural Science Foundation for Young Scholars (2208085QB41), and the Fellowship of China Postdoctoral Science Foundation (2021M693058).

## Author contributions

J.W., Z.Z., and J.Z. designed the study. J.W. and H.T. performed most of the experiments and analyzed the experimental data. L.S. carried out DFT calculations. R.W. provided help with the XAFS measurements. J.W., M.F., J.-L.W., and Y.W. conducted characterizations and analyzed the results. Z.Z. and S.Z. helped in the analysis of data. J.W. and J.Z. wrote the manuscript. All authors discussed the results and assisted during manuscript preparation.

## Competing interests

The authors declare no competing interests.
