## [Peer Review File · Nature Communications]

REVIEWER COMMENTS

Reviewer #1 (Remarks to the Author):

In this study, the authors successfully implemented a precise electrochemical deposition strategy to effectively regulate the site-specific metal-support interactions (MSI) of Ir single-atom catalysts (SACs). By anchoring Ir single atoms on specific sites, particularly at three-fold center cubic hollow sites, stronger MSI was achieved compared to the vacancy site, leading to improved catalytic activity. The authors provided supporting data to validate their proposal. However, similar results have been reported (Adv. Mater. 2018, 30, 1709279; J. Am. Chem. Soc. 2020, 142, 16, 7425–7433), and the electrocatalytic performance after loading the noble metal is not attractive compared to previous results. There are several major concerns.

1. As Ir is very expensive and the global output is pretty low, many works have been proposed to replace the use of Ir. Especially for the alkaline OER, many non-noble metal catalysts have been verified to exhibit very good activity and stability, for example, the NiFe LDHs. So what is the advantage of Ir₁/Ni LDH-T for its application?
2. It is essential to elucidate why Ir⁴⁺ occupies the three-fold center cubic hollow sites, not other locations. To address this, the authors should include detailed electrochemical deposition curves and provide additional discussion in the manuscript. These additions will enhance the understanding of the deposition mechanism and shed light on the preferential occupation of Ir⁴⁺ at the specific sites of interest.
3. It was proposed that Ir acted as active sites in Ir₁/Ni LDH-T while Ni acted as active sites in Ir₁/Ni LDH-V. How to understand the active sites are much more than Ir₁/Ni LDH-V and Ni LDH from the ECSA data? Does that mean the high activity partly originates from the high loading of Ir in Ir₁/Ni LDH-T?
4. How many Ir SACs were successfully located on the Ni LDH for Ir₁/Ni LDH-T and Ir₁/Ni LDH-V? The ICP-MS data is suggested to be added. And shall we control the loading mass of Ir in this electrochemical deposition method?
5. Would the Ir SACs dissolve in electrolytes? Is it still stable enough under a larger current density, for example, 500 mA cm⁻²?
6. In-situ Raman spectra were collected in this work to illustrate the active sites. More experimental details are suggested to be added, for example, the structure of the homemade cell, the composition of the electrolyte, and how the catalysts were labeled with ¹⁸O. A reference peak is strongly suggested to be added when conducting this experiment, for example, ClO₄⁻ could be added to the electrolyte, to confirm the positive/negative shift of $\delta(\text{Ni}^{3+}-\text{O})$ is originated from the reaction. Please refer to this work: Nature 600, 81–85 (2021).

Reviewer #2 (Remarks to the Author):

This work deals with the influence of metal-support interaction on the resulting OER performance, using Ir and Ni LDH as a model system. It appears very intriguing that a small difference in processing (cathodic deposition vs. anodic deposition of Ir) might result in a substantial variety of final catalytic properties. Although we appreciate the authors' efforts for this study, the following major issues prevent us from the strong recommendation of the manuscript for publication, though.

1. First, as clarified in Supplementary Figures 9 and 10, the active surface area of Ir₁/Ni LDH-T (~24 mF/cm²) is six-fold larger than that of Ir₁/Ni LDH-V and Ni LDH (~6 mF/cm²). Consequently, the claim of the notable OER activity enhancement by more than 20 folds on the basis of the geometric area of an electrode in Figure 2a is exaggerated too much. A precise comparison between the three samples is shown in Supplementary Figure 10 rather than Figure 2a in the main text. It appears that no dramatic difference in OER activity is observed between Ir₁/Ni LDH-V and Ir₁/Ni LDH-T, if their surface areas are considered.

2. It should be noted that the Ni states of the supports of Ir₁/Ni LDH-T and Ir₁/Ni LDH-V completely differ, as the two samples underwent cathodic and anodic treatment, respectively. In this respect, the Raman results, merely showing Ni³⁺-O peaks only, may be preliminary and too simplified in order for the authors to strongly claim the exclusion of Ni as active sites in Ir₁/Ni LDH-T. As the support (Ni LDH) is a well-known highly efficient OER catalyst, the contribution of the support to the overall OER activity should not be overlooked.

3. We note that the OER activity of Ir₁/Ni LDH-V is higher than that of Ni LDH. Consequently, Ir is another active site even in Ir₁/Ni LDH-V, in contrast to the authors' claim in Figure 4 by the DFT calculations.

4. One of the most important features in this work is the distinct geometries of Ir between the two samples. We believe that the precise determination of these two discriminated geometries is very critical for DFT calculations. Although the authors attempted to show the distinct configurations of Ir via EXAFS in Figure 1, unfortunately, there is no significant difference between the two curves in Figure 1g. We think that this is one of the most deficiencies to support the claims of the authors.

5. Readers would wonder how the mass of Ir deposited on the support was measured. Without the information on the precise mass, the mass activity and TOF with respect to Ir in Figure 2c would be seriously inaccurate.

Reviewer #3 (Remarks to the Author):

This paper reports on catalyst materials that use varied Ir electrodeposition parameters to produce single atoms of Ir (which become oxidized – IrO_x) / Ni layered double hydroxides. A cathodic deposition of Ir results in a more active catalyst compared to the anodic Ir deposition; the authors identify the strength of MSI as the reason for this activity difference. The claimed specificity of electrodeposition of single atoms onto distinct sites is indeed interesting and a unique, powerful approach for catalyst tuning. However, a major limitation of this work is the lack of strong direct evidence for this claim (both the single atom nature of the Ir and the ability to deposit only in these specific sites). Without such evidence, it is unclear whether these claims are substantiated and what the impact of this work is in its current form. There are also other concerns with the presented conclusions and results. Several questions and suggestions for improvement of this work are detailed below:

1. The authors provide very specific overpotential values for the two catalysts with no error bars or uncertainty range provided. The following sentence is on page 3: “We found that Ir₁/Ni LDH-T exhibited an OER overpotential of 228 mV at 10 mA cm⁻², which was 73 mV lower than Ir₁/Ni LDH-V.” The precision of these values should be reviewed and reflected with appropriate error bars.
2. Authors specify Ni LDH are nanosheets that are “about 3.6 nm.” However, this size figure seems to only be determined from a single nanosheet, while others look to be ~7-13 nm in height/length based on the AFM data provided. This reviewer suggests to simply use the observed range for nanosheet height/length description.
3. Authors mention “metal support interactions” frequently throughout the text, describing some coordination environments of Ir as having “stronger MSI.” However, the authors do not explicitly offer a definition of exactly what is meant by metal support interactions. This is especially unclear since the Ir is oxidized before/during use as a catalyst for water oxidation, such that this material does not exhibit typical metal-support interactions. Do the authors mean that “stronger MSI” materials have a stronger or more covalent bond between Ir sites and coordinated oxygen? Do they mean bond length to nearest neighbor is shortened? An explicit, physical meaning of “stronger” or “weaker” MSI would be helpful to the reader. This physical meaning should then coincide with the geometric environment characterization completed by the authors.
4. In the manuscript, the authors claim “In the case of Ir₁/Ni LDH-T, IrCl₃⁺ cations were deposited on the three-fold fcc hollow site of oxygen, followed by electrooxidation activation to form IrO₆ octahedra linked to Ni LDH support. As for Ir₁/Ni LDH-V, IrO₆ octahedra was directly connected to oxygen vacancy sites on Ni LDH support.” However, it is unclear how this conclusion was made, with no explicit experimental evidence of characterization provided to identify the exact deposition site of Ir within the

support. This is a significant concern for the conclusions made in this paper. Further explanation and evidence is needed to support this claim.

Furthermore, the single atom nature of the Ir is questionable; the provided TEM does show some bright spots that appear to be single atoms, but also shows more clustered areas, even within the small field of view of the provided images. Stronger evidence should be provided to support the single atom nature, or perhaps this should be rephrased to indicate that there are likely some single atoms and some clusters of Ir.

5. The authors make several references to IrCl₃ + species, which is quite unexpected to see this as an ion species; the more common, stable compound is iridium(III) chloride (IrCl₃). The authors do not provide any evidence for presence of Cl, other than identifying an EXAFS peak (which does not serve to identify elements, only to identify neighbors and varying distances). Why do the authors propose that Cl is still present after electrodeposition? Additional data (XPS or XAS) for Cl would be useful to demonstrate presence of Cl.

6. Authors find that Ir/Ni LDH-T has Ni with an oxidized state, after transferring electrons to Ir sites in the material. However, one would expect a reduced Ir state with such a transfer, which is not consistent with the XANES data provided, which shows a highly oxidized Ir state. While this reviewer agrees that a more oxidized Ni is reasonable for Ir/Ni LDH-T based on the data provided, there is currently no provided evidence that an electron transfer to local Ir occurs. Authors should adjust this finding or provide further explanation/evidence of this claimed phenomenon.

7. Specification of potential at which EIS is performed in Figure 2.e is needed.

8. The authors should include the amplitude reduction factor (S₀₂) in Supplementary Table 1. Using an accurate S₀₂ is critical to EXAFS analysis, as it correlates perfectly with CN results. Standard EXAFS fitting procedure involves finding S₀₂ at the specific beamline and edge used based on a standard sample with well known coordination. This analysis should be provided or results stated to complement Supplementary Table 1, with the standard sample and measurement mode specified.

9. On Page 8, the authors state, "In other words, the cathodically deposited Ir atoms on the supports could significantly increase the number of active sites, while the anodically deposited Ir atoms had no obvious effect on the active sites."

Authors use ECSA here as an analog to number of active sites, which is not necessarily appropriate (especially since the ECSA measurements completed are not Ni or Ir site specific, and both samples have

similar Ir/Ni compositions). This reviewer recommends the authors change the mention “actives sites” to simply ECSA.

10. The LSV activity data in Figure 2a and Supplementary Figure 10 curves back on itself (red data in both, around 1.5 V / above 600 mA/cm² in Fig 2a and around 1.5 V / just above 1.5 mA/cm² in Supp Fig 10) indicating that this data is likely overcorrected for iR drop (using too high of a series resistance) making this data look too good, and in fact, unphysical. The authors should check to make sure that no data in the paper is overcorrected, and certainly fix this example.

11. Minor Comments:

a. Typo--This reviewer believes the authors mean “assess” instead of “access” in “To clarify the intrinsic activity of the samples, we tested electrochemical double-layer capacitance (Cdl) to access electrochemically active surface area (ECSA)”.

b. This reviewer finds the authors’ use of past and conditional tenses to be confusing, particularly in the introduction. This reviewer suggests that readers adjust the tenses used to be present tense to strengthen the clarity of scientific language used.

Reviewer #4 (Remarks to the Author):

Point-by-point response to reviewers' comments

Manuscript ID: NCOMMS-23-23885

MS Type: Article

Title: "Site-specific metal-support interaction to switch the activity of Ir single atoms for oxygen evolution reaction"

Reviewer #1

In this study, the authors successfully implemented a precise electrochemical deposition strategy to effectively regulate the site-specific metal-support interactions (MSI) of Ir single-atom catalysts (SACs). By anchoring Ir single atoms on specific sites, particularly at three-fold center cubic hollow sites, stronger MSI was achieved compared to the vacancy site, leading to improved catalytic activity. The authors provided supporting data to validate their proposal. However, similar results have been reported (*Adv. Mater.* **2018**, *30*, 1709279; *J. Am. Chem. Soc.* **2020**, *142*, 16, 7425-7433), and the electrocatalytic performance after loading the noble metal is not attractive compared to previous results. There are several major concerns.

Response: We sincerely thank this reviewer for his/her valuable comments to help us improve the quality of our manuscript. Despite that single-atom Ir catalysts have been reported in previous works, the key point in our work is aiming at the investigation into the specific-site MSI of single atoms towards OER. In previous works, single-atom Ir catalysts were usually synthesized by conventional liquid-phase reduction or thermal treatment (*Adv. Mater.* **2018**, *30*, 1707522; *J. Am. Chem. Soc.* **2020**, *142*, 7425). These synthetic methods could not satisfy the demand for the investigation of the site-specific single atoms towards OER. In our work, we employed a unique and facile electrochemical deposition strategy to construct single-atom Ir catalysts, which is more applicable for investigating site-specific MSI. The positive and negative electric fields can selectively drive precursor ions with different electrical charges, thereby enabling the precise anchoring of Ir single atoms on specific sites of Ni LDH.

In view of the catalytic performance, Ir₁/Ni LDH-T exhibited an OER overpotential of 228 mV at 10 mA cm⁻², which was comparable with the reported results (CoIr-0.2, 235 mV; Ir-NiO, 215 mV). It is worth noting that the content of Ir in Ir₁/Ni LDH-T was only 2.2 wt%, which was much lower than that in CoIr-0.2 (9.7 wt%) and Ir-NiO (18 wt%), greatly decreasing the cost for the catalyst. In our work, the enhanced OER activity of Ir₁/Ni LDH-T was attributed to the strong MSI between Ir single atoms and Ni LDH, optimizing the adsorption strength of intermediates and inducing the switch of the catalytic active sites from Ni sites to Ir sites. This work provides a novel strategy to design highly active catalysts with site-specific MSI by electrochemically selective anchoring of single atoms and offers an in-depth understanding of the correlation between MSI and OER performance.

1. As Ir is very expensive and the global output is pretty low, many works have been proposed to replace the use of Ir. Especially for the alkaline OER, many non-noble metal catalysts have been verified to exhibit very good activity and stability, for example, the NiFe

LDHs. So what is the advantage of Ir₁/Ni LDH-T for its application?

Response: We sincerely thank the reviewer for his/her valuable comments. Despite the high cost of Ir, the content of Ir in Ir₁/Ni LDH-T is really low (2.2 wt%), greatly decreasing the cost of the whole catalyst. Ir₁/Ni LDH-T with such low content of Ir exhibited a remarkable activity with an overpotential of 228 mV at 10 mA cm⁻², which significantly outperformed NiFe LDH (306 mV) (*Nat. Commun.* **2022**, *13*, 2191). Moreover, Ir₁/Ni LDH-T achieved a mass activity of 0.33 A mg⁻¹ at the overpotential of 300 mV, which was 11 times higher than NiFe LDH mentioned above (0.03 A mg⁻¹), indicating the superior activity of Ir₁/Ni LDH-T. The enhanced OER activity of Ir₁/Ni LDH-T was attributed to the strong MSI between Ir single atoms and Ni LDH, optimizing the adsorption strength of intermediates and inducing the switch of the catalytic active sites from Ni sites to Ir sites. A unique and facile electrochemical deposition strategy was employed in this work to precisely anchor the Ir atoms on the specific site of the supports, which was applicable for investigating site-specific MSI.

2. It is essential to elucidate why Ir⁴⁺ occupies the three-fold center cubic hollow sites, not other locations. To address this, the authors should include detailed electrochemical deposition curves and provide additional discussion in the manuscript. These additions will enhance the understanding of the deposition mechanism and shed light on the preferential occupation of Ir⁴⁺ at the specific sites of interest.

Response: We sincerely thank the reviewer for his/her constructive comments. As suggested, we have provided the electrochemical deposition curves as shown in **Figure R1 (Supplementary Fig. 3)**. In the electrochemical cathodic deposition, the current gradually increased with the increase of deposition cycles after the introduction of IrCl₄ as the precursor. The increased cathodic current was derived from the activity of Ir single atoms towards hydrogen evolution, suggesting the successful anchoring of Ir atoms on Ni LDH support. In the case of electrochemical anodic deposition, the slight increase of the current after 5 deposition cycles was consistent with the limited improvement in the OER activity of Ir₁/Ni LDH-V compared with Ni LDH.

Figure R1 (Supplementary Fig. 3). The electrochemical deposition curves of (a) Ir₁/Ni

LDH-T and (b) Ir₁/Ni LDH-V with 100 μM IrCl₄.

In the process of electrochemical deposition, the Ir-based ions are induced onto Ni LDH through electrostatic adsorption. Specifically, under the driving force of electrode potentials, the negative or positive electric field will preferentially induce the deposition of Ir-based cations or Ir-based anions from the solution onto the electrode, respectively. When the defective Ni LDH is employed as the support, it provides different possible sites to deposit single atoms. **Figure R2 (Supplementary Fig. 9)** shows the different possible sites on defective Ni LDH for anchoring single atoms, including the three-fold facial center cubic (*fcc*) hollow site of oxygen, three-fold hexagonal close-packed (*hcp*) hollow site of oxygen, and oxygen vacancy site. The surface oxygen atoms of the three-fold hollow sites provide lone-pair electrons and negative charge to bind Ir-based cations, whereas the deficiency of oxygen induce localized positive charge to combine Ir-based anions.

Figure R2 (Supplementary Fig. 9). Possible anchoring sites of Ni LDH. Three-fold (3-fold) *fcc* hollow site (a), 3-fold *hcp* hollow site (b), and oxygen vacancy site (c). The red and gray spheres represent O and Ni atoms, respectively.

In the process of cathodic deposition to prepare Ir₁/Ni LDH-T, the deposited species in an alkaline environment should be [Ir^{IV}Cl_x(OH)_{3-x}]⁺ (1 ≤ x ≤ 3) cations originated from the IrCl₄ precursor. The valence state of Ir in the deposited species was determined to be +4 by Ir L₃-edge XANES spectra, which showed the similar intensity of white line with IrO₂ for Ir₁/Ni LDH-T before activation (**Supplementary Fig. 8**). Moreover, the fitting results for Ir₁/Ni LDH-T before activation indicated the first-shell coordination of IrCl_{1.4}O_{4.4}, since OH⁻ ions in alkaline environment were prone to attack [IrCl₃]⁺ in the solution or IrCl₃O₃ on Ni LDH (*Radiochim. Acta* **1973**, 20, 47-50). To elucidate why Ir atoms occupy the three-fold center cubic hollow sites, we have conducted DFT calculations to estimate the formation energy (ΔE) of [IrCl₃]⁺, [IrCl₂(OH)]⁺, and [IrCl(OH)₂]⁺ on the surface sites of Ni LDH, respectively (**Fig. R3** or **Supplementary Fig. 11**). Notably, the formation energy of IrCl₃⁺ on the 3-fold *fcc* hollow site is -4.97 eV, which is lower than those on the 3-fold *hcp* hollow site (-3.12 eV) and the oxygen vacancy site (0 eV). Similar results were obtained with [IrCl₂(OH)]⁺ and [IrCl(OH)₂]⁺ as the deposited species, exhibiting the lowest formation energy on the 3-fold *fcc* hollow site. The higher formation energy of [Ir^{IV}Cl_x(OH)_{3-x}]⁺ cations on the *hcp* site may be caused by electrostatic repulsion between inner-layer Ni atom and Ir atoms due to the

closer distance, leading to lower stability. Therefore, the Ir atom of Ir₁/Ni LDH-T was inclined to anchor on the 3-fold *fcc* hollow site.

Figure R3 (Supplementary Fig. 11). Formation energies (ΔE) of $[\text{IrCl}_3]^+$ (a), $[\text{IrCl}_2(\text{OH})]^+$ (b), and $[\text{IrCl}(\text{OH})_2]^+$ (c) anchoring on 3-fold *fcc*, 3-fold *hcp* hollow site, and oxygen vacancy site of Ni LDH (001), respectively. The pink, orange, gray, and purple spheres represent H, Cl, Ni, and Ir atoms, respectively. The red and magenta spheres represent the O atoms bonded to Ni and Ir, respectively.

In the process of anodic deposition to synthesize Ir₁/Ni LDH-V, the negatively charged Ir-based anions were $\text{Ir}(\text{OH})_6^{2-}$ anions with full coordination of IrO_6 due to the alkaline environment (*Nat. Commun.* **2017**, 8, 1341). **Figure R4 (Supplementary Fig. 12)** shows the formation energy of $\text{Ir}(\text{OH})_6^{2-}$ on the 3-fold hollow site and the oxygen vacancy site, respectively. The formation energy of $\text{Ir}(\text{OH})_6^{2-}$ on the oxygen vacancy site (-3.32 eV) is much lower than those on the 3-fold *fcc* (0.18 eV) and *hcp* (0.19 eV) hollow sites, demonstrating the tendency of $\text{Ir}(\text{OH})_6^{2-}$ to anchor on the oxygen vacancy site.

Figure R4 (Supplementary Fig. 12). Formation energies (ΔE) of Ir(OH)_6^{2-} anchoring on 3-fold *fcc* (a), 3-fold *hcp* hollow site (b), and oxygen vacancy site (c) of NiOOH (001). The pink, red, gray, and purple spheres represent H, O, Ni, and Ir atoms, respectively.

Besides, the ions deposited at the cathode and anode have been explored by XPS analysis on the Ir_1/Ni LDH-T before activation and Ir_1/Ni LDH-V, respectively. As shown in **Figure R5**, the Cl $2p$ XPS spectrum of Ir_1/Ni LDH-T before activation exhibited the peaks of Cl $2p_{1/2}$ and $2p_{3/2}$ at 199.6 and 198.2 eV, respectively. The presence of Cl in Ir_1/Ni LDH-T before activation was attributed to $[\text{Ir}^{\text{IV}}\text{Cl}_x(\text{OH})_{3-x}]^+$ ($1 \leq x \leq 3$) cations originated from the IrCl_4 precursor. In comparison, the absence of Cl peak for Ir_1/Ni LDH-V was consistent with the selective deposition of Ir(OH)_6^{2-} after anodic electrochemical deposition.

We have added the corresponding discussions in the revised manuscript (pages 5-6, lines 127-160, highlighted in yellow color).

Figure R5. Cl $2p$ XPS spectra of Ir_1/Ni LDH-T before activation and Ir_1/Ni LDH-V.

3. It was proposed that Ir acted as active sites in Ir₁/Ni LDH-T while Ni acted as active sites in Ir₁/Ni LDH-V. How to understand the active sites are much more than Ir₁/Ni LDH-V and Ni LDH from the ECSA data? Does that mean the high activity partly originates from the high loading of Ir in Ir₁/Ni LDH-T?

Response: We thank the reviewer for raising these issues. In our work, Ir₁/Ni LDH-T and Ir₁/Ni LDH-V have similar Ir/Ni compositions with Ir content of 2.18 wt% and 2.54 wt%, respectively. As such, the high activity of Ir₁/Ni LDH-T did not originate from the high loading of Ir. We are sorry for misleading the reviewer for inappropriately using the measured ECSA as an analog to the number of active sites since the ECSA was not directly related to specific Ir or Ni sites. We have modified the description of ECSA in the revised manuscript.

To investigate the reason for the larger ECSA of Ir₁/Ni LDH-T than Ir₁/Ni LDH-V and Ni LDH, we have analyzed the ECSA of Ni LDH after the same cathodic and anodic electrochemical treatment without the presence of Ir precursors, respectively. As shown in **Figure R6**, the C_{dl} of cathodically treated Ni LDH reached 13.5 mF cm⁻², which was much higher than that of anodically treated Ni LDH (6.4 mF cm⁻²) and pristine Ni LDH (6.0 mF cm⁻²). Likewise, the cathodic electrochemical treatment would increase the ECSA for Ir₁/Ni LDH-T than Ir₁/Ni LDH-V. Besides, we have characterized the morphology of Ni LDH after cathodic and anodic electrochemical treatment, respectively. As displayed in **Figure R7**, small pores appeared for cathodically treated Ni LDH whereas no obvious change was observed for anodically treated Ni LDH. The partial reconstruction of Ni LDH induced by cathodic treatment may lead to the increased ECSA of Ir₁/Ni LDH-T. In addition, Ni LDH, cathodically treated Ni LDH, and anodically treated Ni LDH exhibited similar activity towards OER, implying that the enhanced performance of Ir₁/Ni LDH-T was attributed to the Ir center with site-specific MSI (**Fig. R8** or **Supplementary Fig. 17**).

Figure R6. CV curves and charging current density differences of cathodically treated Ni LDH and anodically treated Ni LDH. CVs for (a) cathodically treated Ni LDH and (b) anodically treated Ni LDH at different scan rates from 20 to 100 mV s⁻¹, respectively. Charging current density differences of (c) cathodically treated Ni LDH and (d) anodically treated Ni LDH, respectively.

Figure R7. TEM images of (a) cathodically treated Ni LDH and (b) anodically treated Ni LDH.

Figure R8 (Supplementary Fig. 17). The polarization curves of Ni LDH, cathodically treated Ni LDH, and anodically treated Ni LDH.

4. How many Ir SACs were successfully located on the Ni LDH for Ir₁/Ni LDH-T and Ir₁/Ni LDH-V? The ICP-MS data is suggested to be added. And shall we control the loading mass of Ir in this electrochemical deposition method?

Response: Thanks for pointing out this. The loadings of Ir for Ir₁/Ni LDH-T and Ir₁/Ni LDH-V were determined to be 2.18 wt% and 2.54 wt% by ICP-AES, respectively. As shown in **Table R1 (Supplementary Table 1)**, we have investigated the loading of Ir with different concentrations of IrCl₄ in the electrolyte. When concentrations of IrCl₄ were set as 25, 50, 100, and 200 μM, the loading of Ir for Ir₁/Ni LDH-T and Ir₁/Ni LDH-V were in the range of 0.5-2.92 wt% and 0.69-3.06 wt%, respectively. It is noteworthy that Ir single atoms aggregated into clusters when the concentration of IrCl₄ was increased to 200 μM (**Fig. R9** or **Supplementary Fig. 7**). We have added the corresponding discussions in the revised manuscript (page 4, lines 99-104, highlighted in yellow color).

Table R1 (Supplementary Table 1). The loading of Ir in Ir₁/Ni LDH-T and Ir₁/Ni LDH-V with different concentrations of IrCl₄.

Concentration of IrCl ₄	Ir ₁ /Ni LDH-T	Ir ₁ /Ni LDH-V
25 μM	0.50 wt%	0.69 wt%
50 μM	1.23 wt%	1.60 wt%
100 μM	2.18 wt%	2.54 wt%
200 μM	2.92 wt%	3.06 wt%

Figure R9 (Supplementary Fig. 7). HAADF-STEM images of Ir₁/Ni LDH-T and Ir₁/Ni LDH-V obtained with a certain concentration of IrCl₄. **a, b** HAADF-STEM images of Ir₁/Ni LDH-T (**a**) and Ir₁/Ni LDH-V (**b**) obtained with 50 μM IrCl₄. **c, d** HAADF-STEM images of Ir₁/Ni LDH-T (**c**) and Ir₁/Ni LDH-V (**d**) obtained with 200 μM IrCl₄. Ir clusters are indicated by red circles.

5. Would the Ir SACs dissolve in electrolytes? Is it still stable enough under a larger current density, for example, 500 mA cm⁻²?

Response: We sincerely thank this reviewer for raising this concern. To test the stability of the catalyst, we have quantified the content of leached Ni and Ir in the electrolyte using ICP-MS after the OER process. As shown in **Figure R10 (Supplementary Fig. 20)**, the percentages of leached Ir and Ni for Ir₁/Ni LDH-T were all less than 5% during the stability test at the current density of 10, 20, and 50 mA cm⁻² for 20 h. Furthermore, after the stability test at a current density of 500 mA cm⁻² for 12 h, the percentages of leached Ir and Ni were 8.2% and 9.6%, respectively (**Fig. R11**). We have added the corresponding discussions in the revised manuscript (page 10, lines 233-235, highlighted in yellow color).

Figure R10 (Supplementary Fig. 20). The percentage of leached Ir and Ni detected by ICP-MS after the stability test at 10, 20, and 50 mA cm⁻² for 20 h, respectively. Leached percentage = (leached mass of Ir or Ni)/(starting mass of Ir or Ni) × 100%.

Figure R11. The percentage of leached Ir and Ni detected by ICP-MS after the test at 500 mA cm⁻² for 12 h. Leached percentage = (leached mass of Ir or Ni)/(starting mass of Ir or Ni) × 100%.

6. In-situ Raman spectra were collected in this work to illustrate the active sites. More experimental details are suggested to be added, for example, the structure of the homemade

cell, the composition of the electrolyte, and how the catalysts were labeled with ^{18}O . A reference peak is strongly suggested to be added when conducting this experiment, for example, ClO_4^- could be added to the electrolyte, to confirm the positive/negative shift of $\delta(\text{Ni}^{3+}\text{-O})$ is originated from the reaction. Please refer to this work: Nature 600, 81-85 (2021).

Response: We genuinely thank this reviewer for his/her valuable suggestions. The photograph of the homemade in situ Raman cell has been added in the revised supplementary information as shown in **Figure R12 (Supplementary Fig. 22)**. A homemade Teflon Raman cell was used for the in situ electrochemical Raman measurements with a 3 mm Au electrode, a Pt wire, and an Ag/AgCl electrode as the working, counter, and reference electrodes, respectively. To label the catalyst with ^{18}O , we conducted LSV measurements in a 1.0 M $\text{KOH-H}_2^{18}\text{O}$ solution within the potential ranging from 0.50 to 1.65 V vs RHE, promoting the exchange of lattice oxygen (*Angew. Chem. Int. Ed.* **2021**, *60*, 3095). Compared with ^{16}O -labelled NiOOH, the two main peaks of ^{18}O -labelled NiOOH shifted to lower wavenumbers by about 20 cm^{-1} due to lattice oxygen exchange, manifesting the occurrence of isotope exchange (**Fig. 3d**).

To further confirm the shift of $\delta(\text{Ni}^{3+}\text{-O})$, we have conducted the in situ Raman test with the addition of 50 mM KClO_4 as an internal standard. As shown in **Figure R13 (Supplementary Fig. 25)**, the peak at 935 cm^{-1} was assigned to the symmetric stretching mode of ClO_4^- ($\nu(\text{ClO}_4^-)$) in the 1.0 M $\text{KOH-H}_2^{16}\text{O}$ solution. During LSV measurement from 0.50 to 1.65 V vs RHE for five cycles in 1.0 M $\text{KOH-H}_2^{18}\text{O}$ solution, the peak of $\nu(\text{ClO}_4^-)$ remained unchanged whereas the peak of $\delta(\text{Ni}^{3+}\text{-O})$ gradually shifted to 457 cm^{-1} . This result indicated that the oxygen in Ni LDH was successfully labeled with ^{18}O .

We have added the corresponding discussions in the revised manuscript (page 11, lines 271-274, highlighted in yellow color).

Figure R12 (Supplementary Fig. 22). The photograph of the homemade in situ Raman cell (a) and the Raman cell during the operation (b).

Figure R13 (Supplementary Fig. 25). **a** In situ Raman spectra of Ni LDH at 1.6 V vs RHE in 1.0 M KOH-H₂¹⁶O and 50 mM KClO₄. **b** In situ Raman spectra of Ni LDH during LSV measurement from 0.50 to 1.65 V vs RHE for five cycles in 1.0 M KOH-H₂¹⁸O and 50 mM KClO₄.

Reviewer #2

This work deals with the influence of metal-support interaction on the resulting OER performance, using Ir and Ni LDH as a model system. It appears very intriguing that a small difference in processing (cathodic deposition vs. anodic deposition of Ir) might result in a substantial variety of final catalytic properties. Although we appreciate the authors' efforts for this study, the following major issues prevent us from the strong recommendation of the manuscript for publication, though.

Response: We genuinely thank this reviewer for his/her valuable comments to help us improve the quality of our manuscript. We have conducted additional characterizations and calculations to strengthen the results in our work. We hope the revised manuscript can provide a clearer picture on the conclusions we draw.

1. First, as clarified in Supplementary Figures 9 and 10, the active surface area of Ir₁/Ni LDH-T (~24 mF/cm²) is six-fold larger than that of Ir₁/Ni LDH-V and Ni LDH (~6 mF/cm²). Consequently, the claim of the notable OER activity enhancement by more than 20 folds on the basis of the geometric area of an electrode in Figure 2a is exaggerated too much. A precise comparison between the three samples is shown in Supplementary Figure 10 rather than Figure 2a in the main text. It appears that no dramatic difference in OER activity is observed between Ir₁/Ni LDH-V and Ir₁/Ni LDH-T, if their surface areas are considered.

Response: We sincerely thank this reviewer for raising this issue. To better clarify the difference in OER activity of Ir₁/Ni LDH-T, Ir₁/Ni LDH-V, and Ni LDH, we have carefully

analyzed the ECSA-normalized LSV curves (**Fig. R14** or **Supplementary Fig. 19**). Remarkably, the overpotential of Ir₁/Ni LDH-T at 1 mA cm⁻² was only 310 mV, which was lower than that of Ir₁/Ni LDH-V (380 mV) and Ni LDH (390 mV). The current density normalized by ECSA for Ir₁/Ni LDH-T was estimated to be 0.48 mA cm⁻² at the overpotential of 300 mV, which was about 5.2-fold and 11.4-fold higher than those for Ir₁/Ni LDH-V (0.093 mA cm⁻²) and Ni LDH (0.042 mA cm⁻²). In this regard, the OER activity of Ir₁/Ni LDH-T normalized by active surface area was still much higher than that of Ir₁/Ni LDH-V and Ni LDH. It is worth noting that the measured ECSA is the average electrochemical active area, which was not directly related to specific Ir or Ni sites. Considering that Ir atoms served as the main active sites in Ir₁/Ni LDH-T, we evaluated the mass activity of the catalysts based on the content of Ir loading. The normalized mass activity of Ir₁/Ni LDH-T at the overpotential of 300 mV reached up to 15.2 A mgr⁻¹, which was 22.7-fold higher than that of Ir₁/Ni LDH-V (**Fig. 2c**). We have added the corresponding discussions in the revised supplementary information.

Figure R14 (Supplementary Fig. 19). ECSA-normalized LSV curves (**a**), overpotentials at the current density of 1 mA cm⁻² (**b**), and the current density normalized by ECSA at an overpotential of 300 mV (**c**) for Ir₁/Ni LDH-T, Ir₁/Ni LDH-V, and Ni LDH, respectively.

2. It should be noted that the Ni states of the supports of Ir₁/Ni LDH-T and Ir₁/Ni LDH-V completely differ, as the two samples underwent cathodic and anodic treatment, respectively. In this respect, the Raman results, merely showing Ni³⁺-O peaks only, may be preliminary and too simplified in order for the authors to strongly claim the exclusion of Ni as active sites in Ir₁/Ni LDH-T. As the support (Ni LDH) is a well-known highly efficient OER catalyst, the contribution of the support to the overall OER activity should not be overlooked.

Response: We appreciate the reviewer's constructive suggestions. To explore the contribution of Ni LDH to OER activity after the electrochemical pretreatment, we have conducted the cathodic and anodic treatment for Ni LDH without the presence of the precursor IrCl₄, respectively. **Figure R15 (Supplementary Fig. 17)** shows the OER performance of Ni LDH, cathodically treated Ni LDH, and anodically treated Ni LDH. The similar polarization curves for the three samples exhibited the overpotentials of 324, 325, and 324 mV at 10 mA cm⁻², respectively, indicating that the activity of Ni LDH after the electrochemical pretreatment remained almost unchanged. Given that Ir₁/Ni LDH-T exhibited remarkably higher activity than Ir₁/Ni LDH-V and Ni LDH, the enhanced performance was attributed to the Ir center

with site-specific MSI rather than Ni LDH. Besides, DFT calculations were performed to clarify the active sites in Ir₁/Ni LDH-T. As shown in **Figure 4g** and **Supplementary Figure 29**, the energy barrier for the rate-determining step (the step from *O to *OOH) on Ir sites in Ir₁/Ni LDH-T is 1.77 eV, significantly lower than that on Ni sites in NiOOH (2.20 eV). This result suggested that Ir atoms preferentially served as the dominant active sites in Ir₁/Ni LDH-T. We have replaced the phrase “true active site” with “dominant active sites” in the revised manuscript (page 12, line 283). We have added the corresponding discussions in the revised manuscript (page 8, lines 203-206, highlighted in yellow color).

Figure R15 (Supplementary Fig. 17). The polarization curves of Ni LDH, cathodically treated Ni LDH, and anodically treated Ni LDH.

3. We note that the OER activity of Ir₁/Ni LDH-V is higher than that of Ni LDH. Consequently, Ir is another active site even in Ir₁/Ni LDH-V, in contrast to the authors’ claim in Figure 4 by the DFT calculations.

Response: Based on the results from DFT calculations, the energy barrier of RDS on Ni sites (2.20 eV) is lower than that on Ir sites (2.34 eV), indicating that the OER process was more favorable on Ni sites for Ir₁/Ni LDH-V (**Fig. 4g** and **Supplementary Fig. 29**). In this case, Ni atoms served as the main active sites in Ir₁/Ni LDH-V. Despite the lower activity of Ir sites compared with Ni sites in Ir₁/Ni LDH-V, the deposition of Ir would provide additional potential active sites for OER, leading to a slight improvement of the activity for Ir₁/Ni LDH-V than Ni LDH.

4. One of the most important features in this work is the distinct geometries of Ir between the two samples. We believe that the precise determination of these two discriminated geometries is very critical for DFT calculations. Although the authors attempted to show the distinct configurations of Ir via EXAFS in Figure 1, unfortunately, there is no significant difference

between the two curves in Figure 1g. We think that this is one of the most deficiencies to support the claims of the authors.

Response: We sincerely acknowledge the reviewer's valuable comments. In the process of electrochemical deposition, the Ir-based ions are induced onto Ni LDH under the driving force of electrode potentials. Specifically, the negative or positive electric field will preferentially induce the deposition of Ir-based cation or Ir-based anions from the solution onto the electrode, respectively. When the defective Ni LDH is employed as the support, it provides different possible sites to deposit single atoms. **Figure R16 (Supplementary Fig. 9)** shows the different possible sites on defective Ni LDH for anchoring single atoms, including the three-fold facial center cubic (*fcc*) hollow site of oxygen, three-fold hexagonal close-packed (*hcp*) hollow site of oxygen, and oxygen vacancy site. The surface oxygen atoms of the three-fold hollow sites provide lone-pair electrons and negative charge to bind Ir-based cations, whereas the deficiency of oxygen induce localized positive charge to combine Ir-based anions.

Figure R16 (Supplementary Fig. 9). Possible anchoring sites of Ni LDH. Three-fold (3-fold) *fcc* hollow site (a), 3-fold *hcp* hollow site (b), and oxygen vacancy site (c). The red and gray spheres represent O and Ni atoms, respectively.

In the process of cathodic deposition to prepare Ir₁/Ni LDH-T, the deposited species in an alkaline environment should be [Ir^{IV}Cl_x(OH)_{3-x}]⁺ (1 ≤ x ≤ 3) cations originated from the IrCl₄ precursor. The valence state of Ir in the deposited species was determined to be +4 by Ir L₃-edge XANES spectra, which showed a similar intensity of white line with IrO₂ for Ir₁/Ni LDH-T before activation (**Supplementary Fig. 8**). Moreover, the fitting results for Ir₁/Ni LDH-T before activation indicated the first-shell coordination of IrCl_{1.4}O_{4.4}, since OH⁻ ions were prone to attack [IrCl₃]⁺ in the solution or IrCl₃O₃ on Ni LDH in alkaline environment (*Radiochim. Acta* **1973**, 20, 47-50). Besides, the existence of Cl in Ir₁/Ni LDH-T before activation has been further confirmed by the Cl 2*p* XPS spectra (**Fig. R17**). To elucidate the specific site for the deposited Ir atoms in Ir₁/Ni LDH-T, we have conducted DFT calculations to estimate the formation energy (ΔE) of [IrCl₃]⁺, [IrCl₂(OH)]⁺, and [IrCl(OH)₂]⁺ on the surface sites of Ni LDH, respectively (**Fig. R18** or **Supplementary Fig. 11**). Notably, the formation energy of IrCl₃⁺ on the 3-fold *fcc* hollow site is -4.97 eV, which is lower than those on the 3-fold *hcp* hollow site (-3.12 eV) and the oxygen vacancy site (0 eV). Similar results

were obtained with $[\text{IrCl}_2(\text{OH})]^+$ and $[\text{IrCl}(\text{OH})_2]^+$ as the deposited species, exhibiting the lowest formation energy on the 3-fold *fcc* hollow site. The higher formation energy of $[\text{Ir}^{\text{IV}}\text{Cl}_x(\text{OH})_{3-x}]^+$ ($1 \leq x \leq 3$) cations on the *hcp* site may be caused by electrostatic repulsion between inner-layer Ni atom and Ir atoms due to the closer distance, leading to lower stability. Therefore, the Ir atom of Ir₁/Ni LDH-T was inclined to anchor on the 3-fold *fcc* hollow site.

Figure R17. Cl 2*p* XPS spectra of Ir₁/Ni LDH-T before activation and Ir₁/Ni LDH-V.

Figure R18 (Supplementary Fig. 11). Formation energies (ΔE) of $[\text{IrCl}_3]^+$ (a), $[\text{IrCl}_2(\text{OH})]^+$ (b), and $[\text{IrCl}(\text{OH})_2]^+$ (c) anchoring on 3-fold *fcc*, 3-fold *hcp* hollow site, and oxygen vacancy site of Ni LDH (001), respectively. The pink, orange, gray, and purple spheres represent H, Cl, Ni, and Ir atoms, respectively. The red and magenta spheres represent the O atoms bonded to Ni and Ir, respectively.

In the process of anodic deposition to synthesize Ir_1/Ni LDH-V, the negatively charged Ir-based anions were $\text{Ir}(\text{OH})_6^{2-}$ anions with full coordination of IrO_6 due to the alkaline environment (*Nat. Commun.* **2017**, 8, 1341). **Figure R19 (Supplementary Fig. 12)** shows the formation energy of $\text{Ir}(\text{OH})_6^{2-}$ on the 3-fold hollow site and the oxygen vacancy site, respectively. The formation energy of $\text{Ir}(\text{OH})_6^{2-}$ on the oxygen vacancy site (-3.32 eV) is much lower than those on the 3-fold *fcc* (0.18 eV) and *hcp* (0.19 eV) hollow sites, demonstrating the tendency of $\text{Ir}(\text{OH})_6^{2-}$ to anchor on the oxygen vacancy site.

We have added the corresponding discussions in the revised manuscript (pages 5-6, lines 127-160, highlighted in yellow color).

Figure R19 (Supplementary Fig. 12). Formation energies (ΔE) of $\text{Ir}(\text{OH})_6^{2-}$ anchoring on 3-fold *fcc* (a), 3-fold *hcp* hollow site (b), and oxygen vacancy site (c) of NiOOH (001). The pink, red, gray, and purple spheres represent H, O, Ni, and Ir atoms, respectively.

5. Readers would wonder how the mass of Ir deposited on the support was measured. Without the information on the precise mass, the mass activity and TOF with respect to Ir in Figure 2c would be seriously inaccurate.

Response: We appreciate the reviewer's constructive suggestions. To quantify the mass of Ir on the electrode surface, the prepared electrode was rinsed several times with ultrapure water and then subjected to ultrasonication in ethanol. The catalyst collected in ethanol was washed three times with ethanol and water by centrifugation. The resulting solid was freeze-dried in a vacuum freeze-dryer overnight. Before analysis using ICP-AES, a proper amount of powder sample was treated by hot aqua regia to get a clear solution. The contents of Ir in Ir_1/Ni LDH-T and Ir_1/Ni LDH-V were measured to be 2.18 wt% and 2.54 wt%, respectively. We have added the corresponding experimental details in the revised manuscript (page 16, lines 383-391, highlighted in yellow color).

Reviewer #3

This paper reports on catalyst materials that use varied Ir electrodeposition parameters to produce single atoms of Ir (which become oxidized – IrO_x)/Ni layered double hydroxides. A cathodic deposition of Ir results in a more active catalyst compared to the anodic Ir deposition; the authors identify the strength of MSI as the reason for this activity difference. The claimed specificity of electrodeposition of single atoms onto distinct sites is indeed interesting and a unique, powerful approach for catalyst tuning. However, a major limitation of this work is the lack of strong direct evidence for this claim (both the single atom nature of the Ir and the ability to deposit only in these specific sites). Without such evidence, it is unclear whether these claims are substantiated and what the impact of this work is in its current form. There are also other concerns with the presented conclusions and results. Several questions and suggestions for improvement of this work are detailed below.

Response: We genuinely thank this reviewer for the insightful questions and valuable suggestions to help us improve the quality of our manuscript. We have added more characterizations to discuss the single-atom nature of Ir. Besides, additional DFT calculations have been also conducted to clarify the ability to deposit on specific sites for Ir-based species. The detailed responses to the reviewer's concerns are listed below.

1. The authors provide very specific overpotential values for the two catalysts with no error bars or uncertainty range provided. The following sentence is on page 3: “We found that Ir₁/Ni LDH-T exhibited an OER overpotential of 228 mV at 10 mA cm⁻², which was 73 mV lower than Ir₁/Ni LDH-V.” The precision of these values should be reviewed and reflected with appropriate error bars.

Response: We acknowledge the reviewer's comments. We have analyzed the overpotential values from three independent measurements. As shown in **Figure R20 (Fig. 2b)**, the overpotential (η) of Ir₁/Ni LDH-T, Ir₁/Ni LDH-V, and Ni LDH was 228 ± 3 mV, 301 ± 8 mV, and 321 ± 3 mV at 10 mA cm⁻². We have updated the corresponding results in the revised manuscript (**Fig. 2b**).

Figure R20 (Fig. 2b). Overpotentials of different catalysts at current densities of 10, 50, and 100 mA cm⁻², respectively. The error bars correspond to the standard deviation of three independent measurements.

2. Authors specify Ni LDH are nanosheets that are “about 3.6 nm.” However, this size figure seems to only be determined from a single nanosheet, while others look to be ~7-13 nm in height/length based on the AFM data provided. This reviewer suggests to simply use the observed range for nanosheet height/length description.

Response: We appreciate the reviewer’s valuable comment. As shown in **Figure R21 (Supplementary Fig. 2)**, we have measured the thickness of multiple nanosheets within the field of view. The thickness of the nanosheets was measured to be 3-10 nm. The corresponding description has been appropriately modified in the revised manuscript (page 4, line 88, highlighted in yellow color).

Figure R21 (Supplementary Fig. 2). AFM image (a) and the corresponding height profiles of Ni LDH nanosheets (b-f).

3. Authors mention “metal support interactions” frequently throughout the text, describing some coordination environments of Ir as having “stronger MSI.” However, the authors do not explicitly offer a definition of exactly what is meant by metal support interactions. This is especially unclear since the Ir is oxidized before/during use as a catalyst for water oxidation, such that this material does not exhibit typical metal-support interactions. Do the authors mean that “stronger MSI” materials have a stronger or more covalent bond between Ir sites and coordinated oxygen? Do they mean bond length to nearest neighbor is shortened? An explicit, physical meaning of “stronger” or “weaker” MSI would be helpful to the reader. This physical meaning should then coincide with the geometric environment characterization completed by the authors.

Response: We genuinely thank this reviewer for his/her valuable comments. In the field of single-atom catalysis, “metal-support interaction” is widely used to describe the interaction or influence between metal active centers and their support materials (*Adv. Mater.* **2020**, *32*, 2003300). MSI is primarily governed by the coordination environment, determined by factors such as the type of coordination (*J. Am. Chem. Soc.* **1995**, *117*, 8407), coordination number (*Nano Res.* **2020**, *13*, 1842), and geometric effects (*Sci. China Mater.* **2020**, *63*, 972). In our work, the stronger MSI in Ir₁/Ni LDH-T was reflected by more covalent bonds between Ir sites and coordinated oxygen from Ni LDH. The Ir atoms in Ir₁/Ni LDH-T were bonded with three oxygen atoms from the three-fold *fcc* hollow site on Ni LDH whereas Ir atoms in Ir₁/Ni LDH-V connected with the oxygen vacancy site through one apex oxygen atom. The stronger MSI between Ir atoms and Ni LDH in Ir₁/Ni LDH-T enhanced the electron transfer from Ni to Ir, which was evidenced by the results of XPS spectra of Ni 2*p* and Ir 4*d* (**Fig. 1h** and **Supplementary Fig. 13**). We have added the corresponding discussions in the revised manuscript (page 6, lines 157-160, highlighted in yellow color).

4. In the manuscript, the authors claim “In the case of Ir₁/Ni LDH-T, IrCl₃⁺ cations were deposited on the three-fold fcc hollow site of oxygen, followed by electrooxidation activation to form IrO₆ octahedra linked to Ni LDH support. As for Ir₁/Ni LDH-V, IrO₆ octahedra was directly connected to oxygen vacancy sites on Ni LDH support.” However, it is unclear how this conclusion was made, with no explicit experimental evidence of characterization provided to identify the exact deposition site of Ir within the support. This is a significant concern for the conclusions made in this paper. Further explanation and evidence is needed to support this claim.

Response: We sincerely acknowledge the reviewer’s valuable comments. In the process of electrochemical deposition, the Ir-based ions are induced onto Ni LDH under the driving force of electrode potentials. Specifically, the negative or positive electric field will preferentially induce the deposition of Ir-based cations or Ir-based anions from the solution onto the electrode, respectively. When the defective Ni LDH is employed as the support, it provides different possible sites to deposit single atoms. **Figure R22 (Supplementary Fig. 9)** shows the different possible sites on defective Ni LDH for anchoring single atoms, including the three-fold (3-fold) facial center cubic (*fcc*) hollow site of oxygen, three-fold hexagonal close-packed (*hcp*) hollow site of oxygen, and oxygen vacancy site. The surface oxygen atoms of the three-fold hollow sites provide lone-pair electrons and negative charge to bind Ir-based cations, whereas the deficiency of oxygen induce localized positive charge to combine Ir-based anions.

Figure R22 (Supplementary Fig. 9). Possible anchoring sites of Ni LDH. Three-fold (3-fold) *fcc* hollow site (a), 3-fold *hcp* hollow site (b), and oxygen vacancy site (c). The red and gray spheres represent O and Ni atoms, respectively.

In the process of cathodic deposition to prepare Ir₁/Ni LDH-T, the deposited species in an alkaline environment should be [Ir^{IV}Cl_x(OH)_{3-x}]⁺ (1 ≤ x ≤ 3) cations originated from the IrCl₄ precursor. The valence state of Ir in the deposited species was determined to be +4 by Ir L₃-edge XANES spectra, which showed a similar intensity of white line with IrO₂ for Ir₁/Ni LDH-T before activation (**Supplementary Fig. 8**). Moreover, the fitting results for Ir₁/Ni LDH-T before activation indicated the first-shell coordination of IrCl_{1.4}O_{4.4}, since OH⁻ ions were prone to attack [IrCl₃]⁺ in the solution or IrCl₃O₃ on Ni LDH in alkaline environment

(*Radiochim. Acta* **1973**, *20*, 47-50). Besides, the existence of Cl in Ir₁/Ni LDH-T before activation has been further confirmed by the Cl 2*p* XPS spectra (**Supplementary Fig. 10**). To elucidate the specific site for the deposited Ir atoms in Ir₁/Ni LDH-T, we have conducted DFT calculations to estimate the formation energy (ΔE) of [IrCl₃]⁺, [IrCl₂(OH)]⁺, and [IrCl(OH)₂]⁺ on the surface sites of Ni LDH, respectively (**Fig. R23** or **Supplementary Fig. 11**). Notably, the formation energy of IrCl₃⁺ on the 3-fold *fcc* hollow site is -4.97 eV, which is lower than those on the 3-fold *hcp* hollow site (-3.12 eV) and the oxygen vacancy site (0 eV). Similar results were obtained with [IrCl₂(OH)]⁺ and [IrCl(OH)₂]⁺ as the deposited species, exhibiting the lowest formation energy on the 3-fold *fcc* hollow site. The higher formation energy of [Ir^{IV}Cl_x(OH)_{3-x}]⁺ cations on the *hcp* site may be caused by electrostatic repulsion between inner-layer Ni atom and Ir atoms due to the closer distance, leading to lower stability. Therefore, the Ir atom of Ir₁/Ni LDH-T was inclined to anchor on the 3-fold *fcc* hollow site.

Figure R23 (Supplementary Fig. 11). Formation energies (ΔE) of [IrCl₃]⁺ (a), [IrCl₂(OH)]⁺ (b), and [IrCl(OH)₂]⁺ (c) anchoring on 3-fold *fcc*, 3-fold *hcp* hollow site, and oxygen vacancy site of Ni LDH (001), respectively. The pink, orange, gray, and purple spheres represent H, Cl, Ni, and Ir atoms, respectively. The red and magenta spheres represent the O atoms bonded to Ni and Ir, respectively.

In the process of anodic deposition to synthesize Ir₁/Ni LDH-V, the negatively charged Ir-based anions were Ir(OH)₆²⁻ anions with full coordination of IrO₆ due to the alkaline environment (*Nat. Commun.* **2017**, *8*, 1341). **Figure R24 (Supplementary Fig. 12)** shows the formation energy of Ir(OH)₆²⁻ on the 3-fold hollow site and the oxygen vacancy site,

respectively. The formation energy of $\text{Ir}(\text{OH})_6^{2-}$ on the oxygen vacancy site (-3.32 eV) is much lower than those on the 3-fold *fcc* (0.18 eV) and *hcp* (0.19 eV) hollow sites, demonstrating the tendency of $\text{Ir}(\text{OH})_6^{2-}$ to anchor on the oxygen vacancy site.

We have added the corresponding discussions in the revised manuscript (pages 5-6, lines 127-160, highlighted in yellow color).

Figure R24 (Supplementary Fig. 12). Formation energies (ΔE) of $\text{Ir}(\text{OH})_6^{2-}$ anchoring on 3-fold *fcc* (a), 3-fold *hcp* hollow site (b), and oxygen vacancy site (c) of NiOOH (001). The pink, red, gray, and purple spheres represent H, O, Ni, and Ir atoms, respectively.

Furthermore, the single atom nature of the Ir is questionable; the provided TEM does show some bright spots that appear to be single atoms, but also shows more clustered areas, even within the small field of view of the provided images. Stronger evidence should be provided to support the single atom nature, or perhaps this should be rephrased to indicate that there are likely some single atoms and some clusters of Ir.

Response: We genuinely thank the reviewer for raising this concern. Since the HAADF-STEM image is a two-dimensional projection, the part overlap of the substrate surface and the high density of single atoms could result in the dense bright spots in the TEM images. The single-atom nature of Ir atoms in Ir_1/Ni LDH-T and Ir_1/Ni LDH-V was further verified by EXAFS analysis, which was a powerful tool for identifying single atoms. As shown in **Figure 2f**, the absence of Ir-Ir bonding at around 2.90 Å excludes the formation of Ir-based clusters or nanoparticles, further confirming the single-atom dispersion of Ir in both samples. In addition, we have explored the formation of Ir clusters by increasing the concentration of IrCl_4 to 200 μM . As shown in **Figure R25 (Supplementary Figure 7c, d)**, the Ir single atoms aggregated into Ir clusters with a size of about 1 nm.

Figure R25 (Supplementary Figure 7c, d). HAADF-STEM images of Ir₁/Ni LDH-T (a) and Ir₁/Ni LDH-V (b) obtained at 200 μ M IrCl₄.

5. The authors make several references to IrCl₃⁺ species, which is quite unexpected to see this as an ion species; the more common, stable compound is iridium(iii) chloride (IrCl₃). The authors do not provide any evidence for presence of Cl, other than identifying an EXAFS peak (which does not serve to identify elements, only to identify neighbors and varying distances). Why do the authors propose that Cl is still present after electrodeposition? Additional data (XPS or XAS) for Cl would be useful to demonstrate presence of Cl.

Response: We appreciate the reviewer's valuable suggestion. In the process of cathodic deposition to prepare Ir₁/Ni LDH-T, the negative electric field will preferentially induce the deposition of Ir-based cation from the solution onto the electrode. In an alkaline environment, the deposited species should be [Ir^{IV}Cl_x(OH)_{3-x}]⁺ (1 ≤ x ≤ 3) cations originated from IrCl₄ precursor. The valence state of Ir in the deposited species was determined to be +4 by Ir L₃-edge XANES spectra, which showed the similar intensity of white line with IrO₂ for Ir₁/Ni LDH-T before activation (**Supplementary Fig. 8**). Moreover, the fitting results for Ir₁/Ni LDH-T before activation indicated the first-shell coordination of IrCl_{1.4}O_{4.4}, since OH⁻ ions in alkaline environment were prone to attack [IrCl₃]⁺ in the solution or IrCl₃O₃ on Ni LDH (*Radiochim. Acta* **1973**, *20*, 47-50). To further confirm the presence of Cl, we have performed XPS analysis on Ir₁/Ni LDH-T before activation and Ir₁/Ni LDH-T. As shown in **Figure R26 (Supplementary Fig. 10)**, the Cl 2p XPS spectrum of Ir₁/Ni LDH-T before activation exhibited Cl 2p_{1/2} and 2p_{3/2} peaks at 199.6 and 198.2 eV, respectively. The presence of Cl was attributed to the deposition species of [Ir^{IV}Cl_x(OH)_{3-x}]⁺ cations originated from the IrCl₄ precursor. After the OER activation, the peak intensity of Cl in Ir₁/Ni LDH-T significantly decreased, indicating oxidative dechlorination during the activation process. We have added the corresponding discussions in the revised manuscript (pages 5-6, lines 136-151, highlighted in yellow color).

Figure R26. Cl 2p XPS spectra of Ir₁/Ni LDH-T before activation and Ir₁/Ni LDH-T.

6. Authors find that Ir/Ni LDH-T has Ni with an oxidized state, after transferring electrons to Ir sites in the material. However, one would expect a reduced Ir state with such a transfer, which is not consistent with the XANES data provided, which shows a highly oxidized Ir state. While this reviewer agrees that a more oxidized Ni is reasonable for Ir/Ni LDH-T based on the data provided, there is currently no provided evidence that an electron transfer to local Ir occurs. Authors should adjust this finding or provide further explanation/evidence of this claimed phenomenon.

Response: We thank the reviewer for the constructive comments. To investigate the electron transfer to the Ir centers, we have carefully analyzed the intensity of the white lines for the XAS of Ir/Ni LDH-T and Ir₁/Ni LDH-T. The enlarged inset in **Figure 2e** shows the relatively weaker intensity of the white line for Ir₁/Ni LDH-T than Ir₁/Ni LDH-V, indicating a lower valence state of Ir in Ir₁/Ni LDH-T. This phenomenon is attributed to the transfer of electrons from Ni LDH to Ir. Moreover, we have conducted Ir 4d XPS measurements to further verify the electron transfer between Ir and Ni LDH (**Fig. R27** or **Supplementary Fig. 13**). The Ir 4d XPS spectra of Ir₁/Ni LDH-T and Ir₁/Ni LDH-V exhibited the characteristic peaks of Ir 4d_{3/2} at 313.5 and 314.0 eV, respectively. The negative shift in the Ir 4d_{3/2} XPS peak of Ir₁/Ni LDH-T suggests a relatively higher electron density associated with Ir single atoms, indicative of enhanced charge transfer from Ni LDH to Ir in Ir₁/Ni LDH-T. We have added the corresponding discussions in the revised manuscript (page 6, lines 165-167, highlighted in yellow color).

Figure R27 (Supplementary Fig. 13). Ir 4d XPS spectra of Ir₁/Ni LDH-T and Ir₁/Ni LDH-V.

7. Specification of potential at which EIS is performed in Figure 2.e is needed.

Response: We genuinely thank the reviewer for raising this issue. EIS measurements were performed at a potential of 1.57 V vs RHE. We have added the corresponding detail in the revised figure caption of **Figure 2e**.

8. The authors should include the amplitude reduction factor (S02) in Supplementary Table 1. Using an accurate S02 is critical to EXAFS analysis, as it correlates perfectly with CN results. Standard EXAFS fitting procedure involves finding S02 at the specific beamline and edge used based on a standard sample with well known coordination. This analysis should be provided or results stated to complement Supplementary Table 1, with the standard sample and measurement mode specified.

Response: We sincerely thank this reviewer for his/her advice. During curve fittings, the amplitude reduction factor S02 was fixed at the value of 0.82 determined by fitting the data of IrO₂. We have made the corresponding revisions in the revised **Supplementary Table 2**.

9. On Page 8, the authors state, “In other words, the cathodically deposited Ir atoms on the supports could significantly increase the number of active sites, while the anodically deposited Ir atoms had no obvious effect on the active sites.” Authors use ECSA here as an analog to number of active sites, which is not necessarily appropriate (especially since the ECSA measurements completed are not Ni or Ir site specific, and both samples have similar Ir/Ni compositions). This reviewer recommends the authors change the mention “active sites” to simply ECSA.

Response: We sincerely thank this reviewer for his/her comments. We have changed the description of “active sites” to “ECSA”. The statement “In other words, the cathodically deposited Ir atoms on the supports could significantly increase the number of active sites, while the anodically deposited Ir atoms had no obvious effect on the active sites.” has been changed to “The C_{dl} value of Ir₁/Ni LDH-T was determined to be 23.4 mF cm⁻², which was higher than those of Ir₁/Ni LDH-V (6.2 mF cm⁻²) and Ni LDH (6.0 mF cm⁻²), indicating an increase in ECSA.”

10. The LSV activity data in Figure 2a and Supplementary Figure 10 curves back on itself (red data in both, around 1.5 V / above 600 mA/cm² in Fig 2a and around 1.5 V / just above 1.5 mA/cm² in Supp Fig 10) indicating that this data is likely overcorrected for iR drop (using too high of a series resistance) making this data look too good, and in fact, unphysical. The authors should check to make sure that no data in the paper is overcorrected, and certainly fix this example.

Response: We sincerely thank this reviewer for raising this concern. The uncompensated ohmic electrolyte resistance (R_u) is extracted from the Nyquist plot via high frequency alternating current impedance (*Chem. Mater.* 2016, 29, 120). In this work, the R_u value was measured to be ~3.0 Ω (1.0 M KOH). We have provided the non-iR corrected data in the revised supplementary information (**Supplementary Fig. 14b**). We have carefully checked the data to make sure that the iR-corrected data were reasonable in this work. The slight backward bending of individual data points in the LSV curve could be attributed to the fluctuation of the LSV curve caused by the generation of numerous bubbles during the high-current OER process.

11. Minor Comments:

a. Typo--This reviewer believes the authors mean “assess” instead of “access” in “To clarify the intrinsic activity of the samples, we tested electrochemical double-layer capacitance (Cdl) to access electrochemically active surface area (ECSA)”.

Response: We thank the referee for the kind suggestion. We have corrected the “access” to “assess” in the revised manuscript (page 9, line 220, highlighted in yellow color).

b. This reviewer finds the authors’ use of past and conditional tenses to be confusing, particularly in the introduction. This reviewer suggests that readers adjust the tenses used to be present tense to strengthen the clarity of scientific language used.

Response: We sincerely thank this reviewer for pointing out the language issue. We have adjusted the language tense according to the reviewer’s suggestion in the revised manuscript.

Reviewer #4

I co-reviewed this manuscript with one of the reviewers who provided the listed reports. This is part of the Nature Communications initiative to facilitate training in peer review and to

provide appropriate recognition for Early Career Researchers who co-review manuscripts.

Response: We sincerely thank this reviewer for his/her careful reading of our manuscript.

REVIEWER COMMENTS

Reviewer #1 (Remarks to the Author):

The authors have addressed my concerns well. The paper can be published as it is.

Reviewer #2 (Remarks to the Author):

It remains somewhat skeptical whether the supercells and the assumptions used in the DFT calculations in this manuscript have reasonably and consistently supported the experimental results. Furthermore, such a 20-fold enhancement of the OER catalysis in Ni LDH for alkaline electrolysis is easily achievable by simply adding 'Fe' as NiFe LDH without any costly 'Ir' from a technological viewpoint. Nevertheless, from a mechanistic viewpoint, we appreciate that this study is a piece of scientifically valuable work that demonstrates the significance of metal-support interactions during electrocatalysis, especially the OER, without unrealistic emphasis on the catalysis performance. In this respect, we believe that the data presentation in Figures 2a, 2b, and 2c should be much more reasonable on the basis of 'ECSA normalization', instead merely of the geometric area (Figures 2a and 2b) and the very uncertain Ir mass (Figure 2c), as already shown in Supplementary Figure S19 in the revised version. We recommend this manuscript for publication in Nature Communications.

Reviewer #3 (Remarks to the Author):

The authors have very thoroughly considered and responded to the majority of the reviewer comments. Any lack of further experimental evidence that may have been requested is acknowledged to be beyond feasibility of currently available techniques. However, I would still like to see two additional, small amendments to follow up on previously made comments.

Regarding previous comment #3 from Reviewer #3, I would recommend that the authors offer an explicit definition in the manuscript text of MSI and how they are using it in this study, specifically noting that they equate it to stronger covalency between Ir and coordinated oxygen in Ni LDH. Otherwise, the current description which notes "interactions" as stronger or weaker is quite vague for broad readership in the catalysis community.

Finally, regarding previous comment #6 from Reviewer #3, I would strongly recommend that the authors soften their language describing the results from the Ir XPS data. The XPS data shows a very weak and noisy signal such that a “firm conclusion” of a peak shift is not well supported. The authors could increase the scan numbers or try taking scans at different spots to get better signal or error bars for this result. Alternatively, providing a calculated “difference spectrum” may also be helpful to “prove” or visualize this difference they are trying to highlight. Similarly, the white line intensity from the Ir XANES data (Fig 1e) is very similar between the two materials. This data is not convincing that such a small white line increase suggests real differences in Ir oxidation state. Overall, the authors could soften this claim and/or acknowledge that this oxidation state change is extremely small, if they would like to address this detail in their analysis.

Point-by-point response to reviewers' comments

Manuscript ID: NCOMMS-23-23885A

MS Type: Article

Title: “Site-specific metal-support interaction to switch the activity of Ir single atoms for oxygen evolution reaction”

Reviewer #1

The authors have addressed my concerns well. The paper can be published as it is.

Response: We sincerely thank the reviewer for his/her positive comments.

Reviewer #2

It remains somewhat skeptical whether the supercells and the assumptions used in the DFT calculations in this manuscript have reasonably and consistently supported the experimental results.

Response: We sincerely thank the reviewer for his/her valuable comments. The calculation models were built on two-layer $3 \times 3 \times 3$ supercells to simulate the structure of Ir₁/Ni LDH-T and Ir₁/Ni LDH-V. In addition, the anchoring sites of Ir on the two catalysts were reasonably established based on the mechanism of electrochemical deposition and the results of structural characterization. Although the structure of models could not be perfectly identical with the real structure of the catalysts, the models could be used to simulate the electronic structure and the reaction process. Specifically, the Bader charge of Ir in Ir₁/Ni LDH-T is lower than that in Ir₁/Ni LDH-V, indicating a lower oxidation state of Ir in Ir₁/Ni LDH-T, which was coincident with the results of XPS. Besides, the lower energy barrier of RDS for Ir₁/Ni LDH-T than Ir₁/Ni LDH-V indicated the higher OER activity of Ir₁/Ni LDH-T, which was in agreement with the results obtained from the electrochemical test. Accordingly, the results of DFT calculation consistently supported our experimental results from a theoretical point of view.

Furthermore, such a 20-fold enhancement of the OER catalysis in Ni LDH for alkaline electrolysis is easily achievable by simply adding ‘Fe’ as NiFe LDH without any costly ‘Ir’ from a technological viewpoint. Nevertheless, from a mechanistic viewpoint, we appreciate that this study is a piece of scientifically valuable work that demonstrates the significance of metal-support interactions during electrocatalysis, especially the OER, without unrealistic emphasis on the catalysis performance. In this respect, we believe that the data presentation in Figures 2a, 2b, and 2c should be much more reasonable on the basis of ‘ECSA normalization’, instead merely of the geometric area (Figures 2a and 2b) and the very uncertain Ir mass (Figure 2c), as already shown in Supplementary Figure S19 in the revised version. We recommend this manuscript for publication in Nature Communications.

Response: We appreciate the reviewer's valuable comment. As suggested, the ECSA-normalized LSV curves, overpotentials at the current density of 1 mA cm^{-2} , and the ECSA-normalized current density at an overpotential of 300 mV have been presented in the revised Figures 2a, 2b, and 2c, respectively. We have also revised the corresponding discussions in the revised manuscript (page 9, lines 225-229, highlighted in yellow color).

Reviewer #3

The authors have very thoroughly considered and responded to the majority of the reviewer comments. Any lack of further experimental evidence that may have been requested is acknowledged to be beyond feasibility of currently available techniques. However, I would still like to see two additional, small amendments to follow up on previously made comments.

Response: We sincerely thank the reviewer for his/her positive comments.

Regarding previous comment #3 from Reviewer #3, I would recommend that the authors offer an explicit definition in the manuscript text of MSI and how they are using it in this study, specifically noting that they equate it to stronger covalency between Ir and coordinated oxygen in Ni LDH. Otherwise, the current description which notes "interactions" as stronger or weaker is quite vague for broad readership in the catalysis community.

Response: We acknowledge the reviewer's comments. The stronger MSI in Ir₁/Ni LDH-T was reflected by more covalent bonds between Ir sites and coordinated oxygen from Ni LDH. As suggested, we have offered an explicit definition of MSI in the revised manuscript (page 3, lines 72-77; page 14, lines 332-333; highlighted in yellow color).

Finally, regarding previous comment #6 from Reviewer #3, I would strongly recommend that the authors soften their language describing the results from the Ir XPS data. The XPS data shows a very weak and noisy signal such that a "firm conclusion" of a peak shift is not well supported. The authors could increase the scan numbers or try taking scans at different spots to get better signal or error bars for this result. Alternatively, providing a calculated "difference spectrum" may also be helpful to "prove" or visualize this difference they are trying to highlight. Similarly, the white line intensity from the Ir XANES data (Fig 1e) is very similar between the two materials. This data is not convincing that such a small white line increase suggests real differences in Ir oxidation state. Overall, the authors could soften this claim and/or acknowledge that this oxidation state change is extremely small, if they would like to address this detail in their analysis.

Response: We sincerely appreciate the insightful comments from the reviewer. As suggested, we have increased the scan numbers to thirty scans to improve the quality of XPS data. As shown in **Figure R1 (Supplementary Fig. 13)**, the slight negative shift in the Ir $4d_{3/2}$ XPS peak of Ir₁/Ni LDH-T relative to Ir₁/Ni LDH-V indicated a slightly lower valence state of Ir

atoms, which could be attributed to the mild charge transfer from Ni LDH to Ir in Ir₁/Ni LDH-T. Moreover, as the reviewer mentioned, the difference in white line intensity of the Ir XANES data is somewhat limited. Given the small change in the oxidation state of Ir, we have softened the language used to describe the results obtained from XPS data and XANES data in the revised manuscript.

Figure R1 (Supplementary Fig. 13). Ir 4d XPS spectra scanned for 30 times of Ir₁/Ni LDH-T and Ir₁/Ni LDH-V.